# AUTOREGRESSIVE DIFFUSION MODEL FOR GRAPH GENERATION

## ABSTRACT

Diffusion-based graph generative models have recently obtained promising results for graph generation. However, existing diffusion-based graph generative models are all one-shot generative models that apply Gaussian diffusion in the dequantized adjacency matrix space. Such a strategy can suffer from difficulty in model training, slow sampling speed, and incapability of incorporating constraints. We propose an *autoregressive diffusion* model for graph generation. Unlike existing methods, we define a node-absorbing diffusion process that operates directly in the discrete graph space. For forward diffusion, we design a *diffusion ordering network*, which learns an optimal node absorbing ordering from graph topology. For reverse generation, we design a *denoising network* that uses the reverse node ordering to efficiently reconstruct the graph by predicting one row of the adjacency matrix at a time. Based on permutation invariance of graph generation, we show that the two networks can be jointly trained by optimizing a simple lower bound of data likelihood. Our experiments on six diverse datasets show that our model achieves better or comparable generation performance with previous state-of-the-art, and meanwhile enjoys fast generation speed.

## 1 INTRODUCTION

Generating graphs from a target distribution is a fundamental problem in many domains such as drug discovery (Li et al., 2018), material design (Maziarka et al., 2020), social network analysis (Grover et al., 2019), and public health (Yu et al., 2020). Deep generative models have recently led to promising advances in this problem. Different from traditional random graph models (Erdos et al., 1960; Albert & Barabási, 2002), these methods fit graph data with powerful deep generative models including variational auto-encoders (Simonovsky & Komodakis, 2018), generative adversarial networks (Maziarka et al., 2020), normalizing flows (Madhawa et al., 2019), and energy-based models (Liu et al., 2021). These models are learned to capture complex graph structural patterns and then generate new high-fidelity graphs with desired properties.

Recently, the emergence of probabilistic diffusion models has led to interest in diffusion-based graph generation (Jo et al., 2022). Diffusion models decompose the full complex transformation between noise and real data into many small steps of simple diffusion. Compared with prior deep generative models, diffusion models enjoy both flexibility in modeling architecture and tractability of the model's probability distributions. To the best of our knowledge, there are two existing works on diffusion-based graph generation: Niu et al. (2020) model the adjacency matrices using score matching at different noise scales, and uses annealed Langevin dynamics to sample new adjacency matrices for generation; Jo et al. (2022) propose a continuous-time graph diffusion model that jointly models adjacency matrices and node features through stochastic differential equations (SDEs).

However, existing diffusion-based graph generative models suffer from three key drawbacks: (1) *Generation Efficiency*. The sampling processes of Niu et al. (2020); Jo et al. (2022) are slow, as Niu et al. (2020) requires a large number of diffusion noising levels and Jo et al. (2022) needs to solve a complex system SDEs. (2) *Continuous Approximation*. They convert discrete graphs to continuous state spaces by adding real-valued noise to graph adjacency matrices. Such dequantization can distort the distribution of the original discrete graph structures, thus increasing the difficulty of model training. (3) *Incorporating constraints*. They are both one-shot generation models and hence cannot easily incorporate constraints during the one-shot generation process.

We propose an autoregressive graph generative model named GRAPHARM via *autoregressive diffusion* on graphs. Autoregressive diffusion (ARM) (Hoogeboom et al., 2022a) is an absorbing diffusion process (Austin et al., 2021) for discrete data, where exactly one dimension of the data decays to the absorbing state at each diffusion step. In GRAPHARM, we design *node-absorbing autoregressive diffusion* for graphs, which diffuses a graph directly in the discrete graph space instead of in the dequantized adjacency matrix space. The forward pass absorbs one node in each step by masking it along with its connecting edges, which is repeated until all the nodes are absorbed and the graph becomes empty. We further design a *diffusion ordering network* in GRAPHARM, which is jointly trained with the reverse generator to learn an optimal node ordering for diffusion. Compared with random ordering as in prior ARM models (Hoogeboom et al., 2022a), the learned diffusion ordering not only provides a better stochastic approximation of the true marginal graph likelihood, but also eases generative model training by leveraging structural regularities.

The backward pass in GRAPHARM recovers the graph structure by learning to reverse the node-absorbing diffusion process with a denoising network. The reverse generative process is autoregressive, which makes GRAPHARM easy to handle constraints during generation. However, a key challenge is to learn the distribution of reverse node ordering for optimizing the data likelihood. We show that this difficulty can be circumvented by just using the exact reverse node ordering and optimizing a simple lower bound of likelihood, based on the permutation invariance property of graph generation. The likelihood lower bound allows for jointly training the denoising network and the diffusion ordering network using a reinforcement learning procedure and gradient descent.

The generation speed of GRAPHARM is much faster than the existing graph diffusion models (Jo et al., 2022; Niu et al., 2020). Due to the autoregressive diffusion process in the node space, the number of diffusion steps in GRAPHARM is the same as the number of nodes, which is typically much smaller than the sampling steps in (Jo et al., 2022; Niu et al., 2020). Furthermore, at each step of the backward pass, we design the denoising network to predict one row of the adjacency matrix at one time. The edges to be predicted follow a mixture of Bernoulli distribution to ensure dependencies among each other. This makes GRAPHARM much more efficient than most previous autoregressive graph generation models.

Our key contributions are as follows: (1) To the best of our knowledge, our work is the first *autoregressive* diffusion-based graph generation model, underpinned by a new node self-absorbing diffusion process. (2) GRAPHARM learns an optimal node generation ordering and thus better leverages the structural regularities for autoregressive graph diffusion. (3) We validate our method on both synthetic and real-world graph generation tasks, on which we show that GRAPHARM outperforms existing graph generative models and is efficient in generation speed.

## 2 BACKGROUND

**Diffusion Model and Absorbing Diffusion**    Given a training instance $x_0 \in \mathbb{R}^D$ sampled from the underlying distribution $p_{\text{data}}(x_0)$, a diffusion model defines a forward Markov transition kernel $q(x_t|x_{t-1})$ to gradually corrupt training data until the data distribution is transformed into a simple noisy distribution. The model then learns to reverse this process by learning a denoising transition kernel parameterized by a neural network $p_\theta(x_{t-1}|x_t)$.

Most existing works on diffusion models use Gaussian diffusion for continuous-state data. To apply Gaussian diffusion on discrete data, one can use dequantization method by adding small noise to the data. However, dequantization distorts the original discrete distribution, which can cause difficulty in training diffusion-based models. For example, dequantization on graph adjacency matrices can destroy graph connectivity information and hurt message passing. Austin et al. (2021); Hoogeboom et al. (2021) introduce several discrete state space diffusion models using different Markov transition matrices. Among them, the absorbing diffusion is the most promising one due to its simplicity and strong empirical performance.

**Definition 1** (Absorbing Discrete Diffusion). *An absorbing diffusion is a Markov destruction process defined in the discrete state space. At transition time step $t$, each element $x_t^{(i)}$ in dimension $i$ is independently decayed into an absorbing state with probabilities $\alpha(t)$.*

The absorbing state can be a [MASK] token for texts or gray pixel for images (Austin et al., 2021). The diffusion process will converge to a stationary distribution that has all the mass on the absorbing

state. The reverse of the absorbing diffusion is learned with categorical distribution to generate the original data. Typically, the decaying probabilities $\alpha(t)$ need to be small and the diffusion steps $T$ need to be large to attain good performance.

**Autoregressive Diffusion Model** Although absorbing diffusion directly operates on the discrete data, it still needs a large number of diffusion steps and thus its generation process can be slow. Autoregressive diffusion model (ARM) (Hoogeboom et al., 2022a) describes a fixed absorbing diffusion process where the generative steps is equal to the dimensionality of the data, *e.g.*, the number of pixels in an image or the number of tokens in a piece of text.

**Definition 2** (Autoregressive Diffusion). *An autoregressive diffusion is a stochastic absorbing process where exactly one dimension decays to the absorbing state at a time. The absorbing diffusion is repeated until all the $D$ dimensions are absorbed.*

An equivalent way to describe this process is to first sample a permutation $\sigma \in S_D$, where $S_D$ represents the set of all permutations of the dimension indices $1, \cdots, D$. Then each dimension of the data decays in that order towards the absorbing state. The corresponding generative process then models the variables in the *exact opposite order* of the permutation. ARM amounts to absorbing diffusion with continuous time limit, as detailed in Appendix A.2.

While ARM offers an efficient and general diffusion framework for discrete data, two key questions remains to be addressed for applying ARM for graphs: (1) How do we define absorbing states for inter-dependent nodes and edges in graphs without losing the efficiency of ARM? (2) While ARM imposes a uniform ordering for arriving at an order-agnostic variational lower bound (VLB) of likelihood, a random ordering fails to capture graph topology. How do we obtain a data-dependent ordering that leverages graph structural regularities during generation? In the next section, we address these two challenges in our proposed GRAPHARM model.

## 3 METHOD

A graph is represented by the tuple $G = (V, E)$ with node set $V = \{v_1, \cdots, v_n\}$ and edge set $E = \{(v_i, v_j) | v_i, v_j \in V\}$. We denote by $n = |V|$ and $m = |E|$ the number of nodes and edges in $G$ respectively. Our goal is to learn a graph generative model from a set of training graphs. We assume the graphs are unattributed and focus on graph topology generation.

### 3.1 AUTOREGRESSIVE GRAPH DIFFUSION PROCESS

Due to the dependency between nodes and edges, it is nontrivial to apply absorbing diffusion (Austin et al., 2021) in the discrete graph space. We first define absorbing node state on graphs as follows:

**Definition 3** (Absorbing Node State). *When a node $v_i$ enters the absorbing state, (1) it will be masked and (2) it will be connected to all the other nodes in $G$ by masked edges.*

Instead of only masking the original edges, we connect the masked node $v_i$ to all the other nodes with masked edges as we cannot know $v_i$'s original neighbors in the absorbing state. With the absorbing node state defined, we then need a node decay ordering for the forward absorbing pass. A naïve strategy is to use a random ordering sampled from a uniform distribution as in (Hoogeboom et al., 2022a). In the reverse generation process, the variables will be generated in the exact reverse order, which also follows a uniform distribution. However, such a strategy is problematic for graphs. First, different graph datasets have different structural regularities, and it is key to leverage such regularities to ease generative learning. For example, community-structured graphs typically consist of dense subgraphs that are loosely overlapping. For such graphs, it is an easier learning task to generate one community first and then add the others, but a random node ordering cannot leverage such local structural regularity, which makes generation more difficult. Second, to compute the likelihood, we need to marginalize over all possible node orderings due to node permutation invariance. It will be more sample efficient if we can use an optimized proposal ordering distribution and use importance sampling to compute the data likelihood.

To address this issue, we propose to use a diffusion ordering network $q_\phi(\sigma | G_0)$ such that, at each diffusion step $t$, we sample from this network to select a node $v_{\sigma(t)}$ to be absorbed and obtain the

Figure 1: The autoregressive graph diffusion process. In the forward pass, the nodes are autoregressively decayed into the absorbing states, dictated by an ordering generated by the diffusion ordering network $q_\phi(\sigma|G_0)$. In the reverse pass, the generator network $p_\theta(G_t|G_{t+1})$ reconstructs the graph structure using the reverse node ordering.

corresponding masked graph $G_t$ (Figure 1). This leads to the following definition of our graph autoregressive diffusion process:

**Definition 4** (Autoregressive Graph Diffusion Process). *In autoregressive graph diffusion, the node decay ordering $\sigma$ is sampled from a diffusion ordering network $q_\phi(\sigma|G_0)$ parameterized by $\phi$. Then, exactly one node decays to the absorbing state at a time according to the sampled diffusion ordering. The process proceeds until all the nodes are absorbed.*

The diffusion ordering network follows a recurrent structure $q_\phi(\sigma|G_0) = \prod_t q_\phi(\sigma_t|G_0, \sigma_{(<t)})$. At each step $t$, the distribution of the $t$-th node $\sigma_t$ is conditioned on the original graph $G_0$ and the generated node ordering up to $t-1$, *i.e.*, $\sigma_{(<t)}$. We use a graph neural network (GNN) to encode the structural information in the graph. To capture the partial ordering, we add positional encodings into node features (Vaswani et al., 2017) as in (Chen et al., 2021). We denote the updated node embedding of node $v_i$ after passing the GNN as $\boldsymbol{h}_i^d$, and parameterize $q_\phi(\sigma_t|G_0, \sigma_{(<t)})$ as a categorical distribution:

$$q_\phi(\sigma_t|G_0, \sigma_{(<t)}) = \frac{\exp(\boldsymbol{h}_i^d)}{\sum_{i' \notin V_M} \exp(\boldsymbol{h}_{i'}^d)}, \tag{1}$$

where $V_M$ is the set of node indices that have been masked.

With $q_\phi(\sigma|G_0)$, GRAPHARM can learn to optimize node ordering for diffusion. However, this also requires us to infer the reverse generation ordering in the backward pass. Inferring such a reverse generation ordering is difficult since we do not have access to the original graph $G_0$ in intermediate backward steps. In Section 3.3, we show that it is possible to circumvent inferring this generation ordering by leveraging the permutation invariance of graph generation.

## 3.2 The reverse generative process

In the generative process, a denoising network $p_\theta(G_t|G_{t+1})$ will denoise the masked graph in the reverse order of the diffusion process. We design $p_\theta(G_t|G_{t+1})$ as a graph attention network (GAT) (Veličković et al., 2018; Liao et al., 2019) parameterized by $\theta$, so that the model can distinguish the masked and unmasked edges. For clarity, we use the vallina GAT to illustrate the computing process. However, one can adopt any advanced graph neural network with attentive message passing.

At time $t$, the input to the denoising network $p_\theta(G_t|G_{t+1})$ is the previous masked graph $G_{t+1}$. A direct way is to use $G_{t+1}$ which contains all the masked nodes with their corresponding masked edges. However, during the initial generation steps, the graph is nearly fully connected with masked edges. This has two issues: (1) the message passing procedure will be dominated by the masked edges which makes the messages uninformative. (2) Storing the dense adjacency matrix is memory expensive, which makes the model unscalable to large graphs. Therefore, during each generation step, we only keep the masked node to be denoised with its associated masked edges, while ignoring the other masked nodes. We refer the modified masked graph as $G_t'$, as shown in Figure 2.

The denoising network first uses an embedding layer to encode each node $v_i$ into a continuous embedding space, *i.e.*, $\boldsymbol{h}_i = \text{Embedding}(v_i)$. At $l$-th message passing, we update the embedding of

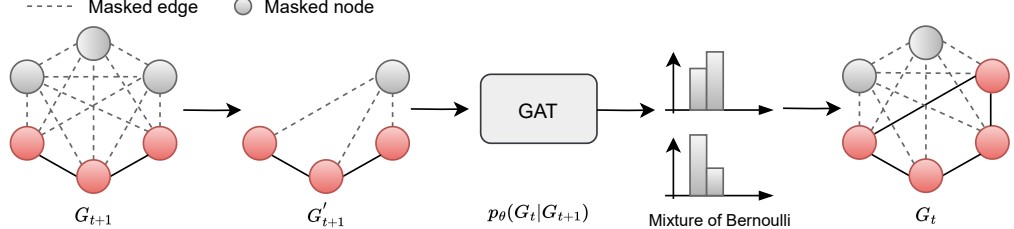

Figure 2: The generation procedure at step $t$ with the denoising network $p_\theta(G_t|G_{t+1})$.

node $v_i$ by aggregating the attentive messages from its neighbor nodes:

$$\alpha_{i,j} = \frac{\exp(\text{LeakyReLU}(\boldsymbol{a}^T[\boldsymbol{W}\boldsymbol{h}_i\|\boldsymbol{W}\boldsymbol{h}_j])}{\sum_{k\in\mathcal{N}_i}\exp(\text{LeakyReLU}(\boldsymbol{a}^T)[\boldsymbol{W}\boldsymbol{h}_i\|\boldsymbol{W}\boldsymbol{h}_j])}, \quad \boldsymbol{h}_i = \text{ReLU}\left(\sum_{j\in\mathcal{N}_i}\alpha_{i,j}\boldsymbol{W}\boldsymbol{h}_j\right), \quad (2)$$

where $\boldsymbol{W}$ is the weight matrix, $\boldsymbol{a}$ is the attention vector. In Eq.2, the attention mechanism enables the model to distinguish if the message comes from a masked edge. After $L$ rounds of message passing, we obtain the final embedding $\boldsymbol{h}_i^L$ for each node, then we predict the edges between the new node $v_{\sigma(t)}$ with all previously denoised nodes $\{v_{\sigma(>t)}\}$. One choice is to sequentially predict these edges as in (You et al., 2018b; Shi et al., 2020). However, this sequential generation process takes $\mathcal{O}(n^2)$ steps, which is inefficient. Instead, we predict the connections of the new node to all previous nodes at once using a mixture of Bernoulli distribution as in (Liao et al., 2019). The mixture distribution can capture the dependencies among edges to be generated and meanwhile reduce the autoregressive generation steps to $\mathcal{O}(n)$.

## 3.3 TRAINING OBJECTIVE

We use approximate maximum likelihood as the training objective for GRAPHARM. We first derive the variational lower bound (VLB) of likelihood as:

$$\begin{aligned}
\log p_\theta(G_0) &= \log\left(\int p(G_{0:n})\frac{q(G_{1:n}|G_0)}{q(G_{1:n}|G_0)}dG_{1:n}\right) \\
&\geq \mathbb{E}_{q(G_{1:n}|G_0)}\left(\log p(G_{1:n}) + \log p(G_0|G_{1:n}) - \log q(G_{1:n}|G_0)\right) \\
&= \mathbb{E}_{q(\sigma_1,\cdots,\sigma_n|G_0)}\sum_t\log p_\theta(G_t|G_{t+1}) - \text{KL}(q(\sigma_1,\cdots,\sigma_n|G_0)|p(\sigma_1,\cdots,\sigma_n|G_n)),
\end{aligned}$$
$$(3)$$

where $G_{0:n}$ denotes all values of $G_t$ for $t = 0,\cdots,n$ and $p(\sigma_1,\cdots,\sigma_n|G_n)$ is the distribution of the generation ordering. A detailed derivation of Eq. 3 is given in Appendix A.3

As we can see from Eq. 3, introducing the diffusion ordering network requires to infer a reverse generation ordering $p(\sigma_1,\cdots,\sigma_n|G_n)$. This is difficult as we do not have the original graph $G_0$ in the intermediate generation process. However, since graph generation is node permutation invariant, we do not need to learn such a generation ordering explicitly. Note that the first term in Eq.3 encourages the model to generate the nodes in the exact reverse ordering of the diffusion process. This pushes the generation ordering $p(\sigma_1,\cdots,\sigma_n|G_n)$ close to the diffusion ordering $q_\phi(\sigma_1,\cdots,\sigma_n)|\tilde{G}_0)$ during training and the KL-divergence term will converge to 0. Therefore, we can ignore the KL-divergence term and finally arrive at a simple training objective:

$$L_{\text{train}} = \mathbb{E}_{\sigma_1,\cdots,\sigma_n\sim q_\phi(\sigma_1,\cdots,\sigma_n|G_0)}\sum_t p_\theta(G_t|G_{t+1}). \quad (4)$$

Compared with the random diffusion ordering, our design has two benefits: (1) We can automatically learn the optimal node generation ordering which leverages the structural information in the graph. (2) We can consider the diffusion ordering network as an optimized proposal distribution of importance sampling for computing the data likelihood, which is more sample-efficient than an uniform proposal distribution.

## 3.4 PARAMETER OPTIMIZATION

Learning the parameters of GRAPHARM is challenging, because we need to evaluate the expectation of the likelihood over the diffusion ordering network. We use a reinforcement learning (RL) procedure by sampling multiple diffusion trajectories, thereby enabling training both the diffusion ordering network $q_\phi(\sigma|G_0)$ and the denoising network $p_\theta(G_t|G_{t+1})$ using gradient descent.

Specifically, at each training iteration, we explore the diffusion ordering network by creating $M$ diffusion trajectories for each training graph $G_0^{(i)}$. Each trajectory is a sequence of graphs $\{G_t^{(i,m)}\}_{1 \le t \le n}$ where the node decay ordering is sampled from $q_\phi(\sigma|G_0)$. The denoising network $p_\theta(G_t|G_{t+1})$ is then trained to minimize the negative VLB using stochastic gradient descent (SGD):

$$\theta_j \leftarrow \theta_{j-1} - \frac{\eta_1}{M} \nabla \sum_{i \in \mathcal{B}_{\text{train}}} \sum_m \sum_t \log p_\theta(G_t^{(i,m)}|G_{t+1}^{(i,m)}), \tag{5}$$

where $\mathcal{B}_{\text{train}}$ is the a minibatch sampled from the training data.

To evaluate the current diffusion ordering network, we create $M$ trajectories for each validation graph and compute the negative VLB of the denoising network to obtain the corresponding rewards $R_t^m = -\sum_{i \in \mathcal{B}_{\text{val}}} \sum_t \log p_\theta(G_t^{(i,m)}|G_{t+1}^{(i,m)})$. Then, the diffusion ordering network can be updated with common RL optimization methods, *e.g.*, the REINFORCE algorithm (Williams, 1992):

$$\phi_j \leftarrow \phi_{j-1} - \frac{\eta_2}{M} \sum_m R_t^m \nabla \log q_\phi(\sigma|G_0). \tag{6}$$

In practice, one can also explore other advanced RL technique, *e.g.*, PPO2, (Schulman et al., 2017) but we find that the simple REINFORCE algorithm has already achieved outstanding performance. The detailed training procedure is summarized in Algorithm 1 in Appendix. A.4.

## 3.5 COMPARISON WITH OM

OM (Chen et al., 2021) also models the node ordering for autoregressive graph generation, but our method differs from OM in two aspects. (1) The motivations are different. Our method is motivated by autoregressive diffusion and treats the graph sequences in the diffusion process as the latent variable; OM treats the node ordering as the latent variable and infers its posterior distribution in a way similar to Variational autoencoder (VAE). (2) Our training objective is much simpler than OM. First, we do not need to compute the complicated graph automorphism, which requires some approximation algorithms to compute. Second, our training objective does not involve the entropy of the node ordering distribution. Existing works (Lucas et al., 2019b;a) have shown that the entropy term in the VAE objective can easily cause posterior collapse.

## 4 EXPERIMENTS

### 4.1 EXPERIMENTAL SETUP

**Dataset**. We evaluate the performance of GRAPHARM on six diverse graph generation benchmarks, covering both synthetic and real-world graphs from different domains: (1) Community-small (You et al., 2018b), (2) Caveman (You, 2018), (3) Cora (Sen et al., 2008), (4) Breast (Gonzalez-Malerva et al., 2011), (5) Enzymes (Schomburg et al., 2004) and (6) Ego-small (Sen et al., 2008). For each dataset, we use $80\%$ of the graphs as training set and the rest $20\%$ as test sets. Following (Liao et al., 2019), we randomly select $20\%$ from the training data as the validation set. We generate the same amount of samples as the test set for each dataset. More details can be seen in Appendix.A.8.

**Baselines**. We compare GRAPHARM with the following baselines: DeepGMG (Li et al., 2018) and GraphRNN (You et al., 2018b) are RNN-based autoregressive graph generation models. GraphAF (Shi et al., 2020) and GraphDF (Luo et al., 2021) are flow-based autoregressive models. GRAN (Liao et al., 2019) is an autoregressive model conditioned on blocks. OM (Chen et al., 2021) is an autoregressive model that infers node ordering using variational inference. GraphVAE (Simonovsky & Komodakis, 2018) is an one-shot model based on VAE. EDP-GNN (Niu et al., 2020) and GDSS (Jo et al., 2022) are scored-based one-shot methods.

| Model | Community-small | | | | Caveman | | | | Cora | | | |
|---|---|---|---|---|---|---|---|---|---|---|---|---|
| | Deg. | Clus. | Orbit | Time/s | Deg. | Clus. | Orbit | Time/s | Deg. | Clus. | Orbit | Time/s |
| DeepGMG | 0.220 | 0.950 | 0.400 | 496.6 | 1.752 | 1.642 | 0.2122 | 530.2 | - | - | - | - |
| GraphRNN | 0.080 | 0.120 | 0.040 | 16.4 | 0.371 | 1.035 | 0.033 | 27.0 | 1.689 | 0.608 | 0.308 | 33.3 |
| GraphAF | 0.180 | 0.200 | 0.020 | 19.3 | 0.269 | 0.587 | 0.422 | 20.5 | 0.176 | 0.080 | 0.094 | 108.5 |
| GraphDF | 0.060 | 0.120 | 0.030 | 10.3 | 0.077 | 0.373 | 0.051 | 18.5 | 0.454 | **0.074** | 0.256 | 129.7 |
| GraphVAE | 0.350 | 0.980 | 0.540 | 0.2 | 1.402 | 1.086 | 1.391 | 0.2 | 1.521 | 1.740 | 0.788 | 0.3 |
| GNF | 0.200 | 0.200 | 0.110 | - | - | - | - | - | - | - | - | - |
| GRAN | 0.060 | 0.110 | 0.050 | 1.8 | 0.043 | 0.130 | 0.018 | 2.5 | 0.125 | 0.272 | 0.127 | 5.1 |
| OM | 0.047 | 0.130 | 0.008 | 2.0 | 0.032 | 0.076 | 0.027 | 3.0 | 0.249 | 0.201 | 0.145 | 5.6 |
| EDP-GNN | 0.053 | 0.144 | 0.026 | 2898.7 | 0.032 | 0.168 | 0.030 | 1823.4 | **0.093** | 0.269 | **0.062** | 4655.1 |
| GDSS | 0.045 | **0.086** | 0.007 | 435.1 | **0.019** | 0.048 | **0.006** | 514.0 | 0.160 | 0.376 | 0.187 | 421.4 |
| GRAPHARM | **0.038** | 0.090 | **0.003** | 1.9 | 0.048 | **0.022** | 0.014 | 2.7 | 0.281 | 0.134 | 0.114 | 4.5 |

| Model | Breast | | | | Enzymes | | | | Ego-small | | | |
|---|---|---|---|---|---|---|---|---|---|---|---|---|
| | Deg. | Clus. | Orbit | Time/s | Deg. | Clus. | Orbit | Time/s | Deg. | Clus. | Orbit | Time/s |
| DeepGMG | - | - | - | - | - | - | - | - | 0.040 | 0.100 | 0.020 | 477 |
| GraphRNN | 0.103 | 0.138 | 0.005 | 31.0 | **0.017** | 0.062 | 0.046 | 19.2 | 0.090 | 0.220 | 0.003 | 18.7 |
| GraphAF | 0.111 | 0.407 | 0.003 | 53.1 | 1.669 | 1.283 | 0.266 | 28.6 | 0.030 | 0.110 | **0.001** | 6.9 |
| GraphDF | 0.283 | 0.078 | 0.035 | 62.4 | 1.503 | 1.061 | 0.202 | 39.8 | 0.040 | 0.130 | 0.010 | 10.2 |
| GraphVAE | 1.591 | 1.993 | 1.050 | 0.2 | 1.369 | 0.629 | 0.191 | 0.7 | 0.130 | 0.170 | 0.050 | 0.3 |
| GNF | - | - | - | - | - | - | - | - | 0.030 | 0.100 | **0.001** | - |
| GRAN | 0.073 | 0.413 | 0.010 | 2.1 | 0.054 | 0.078 | 0.017 | 3.2 | 0.030 | 0.029 | 0.014 | 1.2 |
| OM | 0.042 | 0.140 | 0.005 | 2.8 | 0.051 | 0.083 | 0.024 | 2.4 | 0.024 | 0.035 | 0.018 | 1.4 |
| EDP-GNN | 0.131 | 0.038 | 0.019 | 3694.2 | 0.023 | 0.268 | 0.082 | 2155.5 | 0.052 | 0.093 | 0.007 | 3851.9 |
| GDSS | 0.113 | **0.020** | 0.003 | 858.4 | 0.026 | 0.061 | **0.009** | 443.0 | 0.021 | 0.024 | 0.007 | 260.9 |
| GRAPHARM | **0.031** | 0.046 | **0.002** | 2.3 | 0.033 | **0.059** | 0.019 | 3.5 | **0.016** | **0.014** | 0.012 | 1.2 |

Table 1: Generation results on the six datasets for all the methods. We report the MMD distance between the test datasets and generated graphs. Best results are bold and the second best values are underlined (smaller the better). "-" denotes out-of-resources that take more than 10 days to run.

**Parameter Settings**. We use ADAM with $\beta_1 = 0.9$ and $\beta = 0.999$ as the optimizer. The learning rate is set for $10^{-4}$ and $5 \times 10^{-4}$ for the denoising network and diffusion ordering network respectively on all the datasets. For fair comparison, our denoising network uses the same graph attention network architecture as the baseline GRAN; the diffusion ordering network uses the same graph attention network architecture as the baseline OM. As OM is compatible with any existing autoregressive methods, we use the strongest autoregressive baseline GRAN as its backbone. More implementation details and parameter settings are provided in in Appendix A.6.

**Evaluation metrics**. Following previous work (You et al., 2018b; Jo et al., 2022), we measure generation quality using the maximum mean discrepancy (MMD) as a distribution distance between the generated graphs and the test graphs. Specifically, we compute the MMD of degree distribution, clustering coefficient, and orbit occurrence numbers of 4 nodes between the generated set and the test set. We also report the generation time of different methods.

## 4.2 EXPERIMENTAL RESULTS

**Main Results** Table 1 shows the generation performance on the six benchmarks for all the methods. (1) As we can see, our method can outperform or achieve competitive performance compared with the baselines. In terms of efficiency, our model is on par with the most efficient autoregressive baseline GRAN, and 10-100X faster than other baselines except for the one-shot baseline Graph-VAE. However, GraphVAE's generation quality is much worse than ours as it generates all edges independently from the latent embedding and cannot well capture complex edge dependencies. (2) GDSS is the strongest baseline. However, its generation is extremely slow as it needs to solve a complex SDE system in the reserve process. Our GRAPHARM model is more than 100 times faster than GDSS in generation on almost all the datasets. The other graph diffusion model EDP-GNN is even slower than GDSS as its generation process uses annealed Langevin Dynamics with many different noise levels. (3) GRAPHARM also outperforms existing autoregressive graph generative models by large margins. This is because they adopt a fixed node ordering when training the generative model while GRAPHARM automatically learns an optimal generation ordering for the graph.

Though OM also learns a node ordering distribution, GRAPHARM consistently outperforms it on all the datasets. This is because OM uses the VAE objective which may suffer from posterior collapse, while GRAPHARM has a much simpler training objective based on autoregressive diffusion.

**Constrained Graph Generation** In this set of experiments, we study GRAPHARM's capability in constrained graph generation by comparing it with the strongest one-shot and autoregressive baselines. We use the Caveman dataset for constrained graph generation, with the constraint that the maximum node degree is no larger than 6. The detailed setup is in Appendix A.7. Table. 2 shows the constrained generation performance on the Caveman dataset. GDSS and EDP-GNN are both one-shot generative model and thus they cannot guarantee the constraints are satisfied during sampling. We find that more than half of their generated samples are invalid. For autoregressive baselines, we apply the degree checking procedure on the two strongest baselines, *i.e.*,GRAN and OM. As we can see, GRAPHARM can generate constrained samples that are closer to the data distribution. This is useful for many real-world applications. For example, when designing the contact networks of patients and healthcare workers in hospitals, a constraint of degrees for healthcare workers can help avoid superspreaders and potential infectious disease outbreaks (Jang et al., 2019; Adhikari et al., 2019).

| Method | Validity | Deg. | Clus. | Orbit |
|---|---|---|---|---|
| EDP-GNN | 39% | - | - | - |
| GDSS | 34% | - | - | - |
| GRAN | 100% | 0.208 | 0.231 | 0.158 |
| OM | 100% | 0.190 | 0.168 | **0.132** |
| GRAPHARM | 100% | **0.179** | **0.135** | 0.141 |

Table 2: Constrained graph generation results on the Caveman dataset.

**Evaluation of Log likelihood** We further evaluate the expected negative log-likelihood (NLL) across node permutations on the test sets. For GRAN, we sample 1000 node permutations from the uniform distribution. For OM and GRAPHARM, we sample 1000 node permutations from the ordering network. Table 3 shows the expected NLL on the test sets for Community-small and Breast. As shown, GRAPHARM can achieve competitive results with OM and outperform GRAN by large margins. This is because both OM and GRAPHARM learn an optimal node ordering distribution; sampling from this distribution is more sample efficient than the uniform distribution. Though GDSS and EDP-GNN are also diffusion based graph generative model, they cannot provide the likelihood. Note that GDSS involves a system of SDEs and we cannot directly use the probability flow ODE in (Song et al., 2021) to compute the likelihood. This can be a drawback in some density-based downstream tasks, *e.g.*, outlier detection.

| | Community-small | Breast |
|---|---|---|
| GRAN | 23.04 | 247.18 |
| OM | **16.82** | **187.22** |
| EDP-GNN | N/A | N/A |
| GDSS | N/A | N/A |
| GRAPHARM | **15.79** | **190.34** |

Table 3: Test set negative log-likelihood (NLL) on Community-small and Breast datasets. N/A represents the model cannot provide the NLL.

**Effect of Diffusion Ordering** We further validate the superiority of the learned diffusion ordering in GRAPHARM over random node ordering for graph generation. We compare GRAPHARM with an ablation OA-ARM, which uses random node ordering (Hoogeboom et al., 2022a) for graph generation. Table 4 shows the generation performance and generation ordering of GRAPHARM and OA-ARM on the Community-small dataset. As shown, with the random generation ordering, the generation performance drops significantly. This is because a random generation ordering fails to leverage the structural regularities in the graphs and thus learning the generative model is much more difficult. To evaluate the node generation ordering, we use the spectral graph clustering method to partition the nodes into two clusters for each

| | Deg. | Clus. | Orbit | Counts |
|---|---|---|---|---|
| OA-ARM | 0.159 | 0.372 | 0.291 | 6.9 |
| GRAPHARM | **0.038** | **0.090** | **0.003** | **2.6** |

Table 4: Generation performance and node generation ordering of GRAPHARM and OA-ARM on the Community-small dataset. Counts represents the average number of nodes that cross different clusters during the generation procedure.

generated graph and then count the cross-cluster steps during the generation procedure. As we can see, the average number of nodes that cross different clusters of our method is much smaller than OA-ARM. This demonstrates that our method tends first to generate one cluster and then adds another cluster while OA-ARM just randomly generates the graph. Therefore, the generation ordering of GRAPHARM can better capture graph topology regularity than OA-ARM. Figure 3 (in Appendix A.1) visualize the graph generative process of GRAPHARM and OA-ARM. As we can see, GRAPHARM first generates one community and then moves to another; while OA-ARM randomly generates the graph and fails to capture the underlying graph distribution.

## 5 ADDITIONAL RELATED WORK

**Graph Generation.** *One-shot graph generative models* generate all edges between nodes at once. Models based on VAEs (Simonovsky & Komodakis, 2018; Liu et al., 2018; Ma et al., 2018) and GANs (Cao & Kipf, 2022; Maziarka et al., 2020) generate all edges independently from latent embeddings. This independence assumption can hurt the quality of the generated graphs. Normalizing flow models (Zang & Wang, 2020; Madhawa et al., 2019) are restricted to invertible model architectures for building a normalized probability. The other class is *autoregressive graph generative models*, which generate a graph by sequentially adding nodes and edges. Autoregressive generation can be achieved using recurrent networks (Li et al., 2018; You et al., 2018b; Popova et al., 2019), VAEs (Liu et al., 2018; Jin et al., 2018; 2020), normalizing flows (Shi et al., 2020; Luo et al., 2021), and RL (You et al., 2018a). By breaking the problem into smaller parts, these methods are more apt at capturing complex structural patterns and can easily incorporate constraints during generation. However, a key drawback of them is that their training is sensitive to node ordering. Most existing works pre-define a fixed node ordering by ad-hoc such as random breadth-first search (BFS) ordering (You et al., 2018b; Shi et al., 2020; Luo et al., 2021), which cannot preserve permutation invariance and also be suboptimal for generation.

**Diffusion and Score-Based Generation.** Diffusion models have emerged as a new family of powerful deep generative models. Denoising diffusion probabilistic modeling (DDPM) (Sohl-Dickstein et al., 2015; Ho et al., 2020) perturbs the data distribution into a Gaussian base distribution through an forward Markov noising process, and then learns to recover data distribution via the reverse transition of the Markov chain. Closely related to DDPM is score-based generation (Song & Ermon, 2019), which perturbs data with gradually increasing noise, and then learns to reverse the perturbation via score matching. Song et al. (2021) generalize diffusion models to continuous-time diffusion using forward and backward SDEs. Diffusion and score-based models has been successfully developed for problems including image synthesis (Dhariwal & Nichol, 2021), text-to-image synthesis (Ramesh et al., 2022; Saharia et al., 2022), and molecular conformation modeling (Xu et al., 2021; Hoogeboom et al., 2022b). Few works have studied diffusion-based graph generation. To the best of our knowledge, Niu et al. (2020); Jo et al. (2022) are the only two works in this line. Different from these two works, our model is the first *autoregressive* diffusion model for graph generation, which defines diffusion directly in the discrete graph space.

## 6 LIMITATIONS AND DISCUSSION

We have proposed a new autoregressive diffusion model for graph generation. The proposed model GRAPHARM defines a node-absorbing diffusion process that directly operates in the discrete graph spaces. We designed a diffusion ordering network that learns an optimal ordering for this diffusion process, coupled with a reverse denoising network that performs autoregressive graph reconstruction. We derived a simple variational lower bound of the likelihood, and showed that the two networks can be jointly trained with reinforcement learning. Our experiments have validated the generation performance and efficiency of GRAPHARM.

We discuss several limitations and possible future extensions of GRAPHARM: (1) *Generating attributed graphs.* Our work has been focused on generating graph structures only. However, it is possible to extend GRAPHARM for attributed graph generation by changing the Bernoulli-mixture distribution to multinomial-mixture distribution. (2) *Handling more complex constraints.* We have shown that the autoregressive procedure of GRAPHARM can handle constraints better than one-shot generative models. However, practical graph generation applications can involve complex constraints on global-level properties such as graph spectrum. How to handle such global-level constraints remains challenging for our model and existing graph generative models.

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

# A APPENDIX

## A.1 VISUALIZATION OF GENERATION ORDERING

Figure 3 shows the graph generative process of GRAPHARM and OA-ARM. As we can see, GRAPHARM first generates one community and then moves to another; while OA-ARM randomly generates the graph and fails to capture the underlying graph distribution.

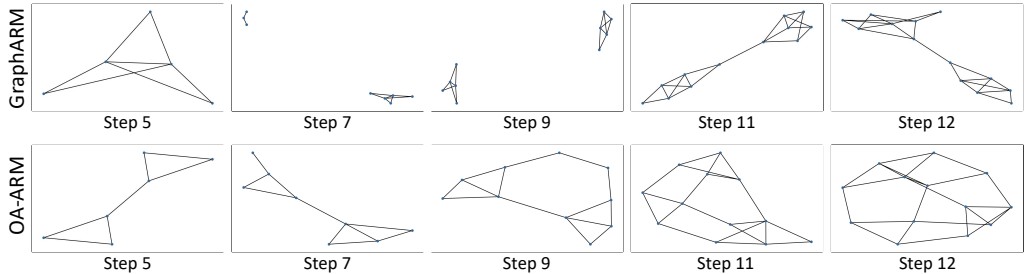

Figure 3: The graph generative process of GRAPHARM and OA-ARM for community generation. As we can see, GRAPHARM first generates one community and then adds another one, which show that GRAPHARM captures graph structural topology for generation. In contrast, OA-ARM generates the graph with a random order.

## A.2 CONNECTION BETWEEN AUTOREGRESSIVE DIFFUSION AND ABSORBING DIFFUSION

While ARM appears different from classic diffusion, it amounts to absorbing diffusion with continuous time limit. Starting from state $\boldsymbol{x}_0$, we can define a continuous-time absorbing process, where each element $\boldsymbol{x}_t^{(i)}$ independently decays into an absorbing state with continuous-time probabilities $\alpha(t)$. This stochastic process is equivalent as using a finite set of $D$ random transition times $\{\tau_i\}_{i=1}^D$ for recording the time where $\boldsymbol{x}_t^{(i)}$ was absorbed. It was shown by Hoogeboom et al. (2022a) that modeling the reverse generation of this process does not need to be conditioned on the precise values of the transition times $\tau_i$. Hence, when training the reverse generative model, we only need to model $\boldsymbol{x}_{\tau_i}$ based on $\boldsymbol{x}_{\tau_{i+1}}$ while ignoring $\tau_i$. This allows for writing the variational lower bound (VLB) of likelihood as an expectation over an uniform ordering in an autoregressive form:

$$\log p(\boldsymbol{x}_0) \geq \mathbb{E}_{\sigma \sim \mathcal{U}(S_d)} \sum_i^D \log p(\boldsymbol{x}_{\sigma(i)}|\boldsymbol{x}_{\sigma(<i)}). \tag{7}$$

## A.3 DERIVATION OF EQ. 3

Following (Chen et al., 2021), let us define $\Pi(G_{1:n})$ as the set of all node orderings that give the same graph sequence $G_{1:n}$, i.e., $\Pi(G_{1:n}) = \{\sigma_{1:n} : G_t(\sigma_{1:t}) = G_t, \forall t = 1, \cdots, n\}$ where $G_t(\sigma_{1:t})$ is the masked graph at time step $t$ under the node ordering $\sigma_{1:t}$. Following (Chen et al., 2021), the conditional $p(\sigma_{1:n}|G_{1:n})$ is a uniform distribution, i.e., $p(\sigma_{1:n}|G_{1:n}) = \frac{1}{|\Pi(G_{1:n})|}, \forall \sigma_{1:n} \in \Pi(G_{1:n})$. Then, we have the following two equations (Chen et al., 2021):

$$p_\theta(G_{1:n}, \sigma_{1:n}) = \frac{1}{|\Pi(G_{1:n})|} p_\theta(G_{1:n}), \tag{8}$$

$$q_\phi(G_{1:n}|G_0) = |\Pi(G_{1:n})| q_\phi(\sigma_{1:n}|G_0). \tag{9}$$

To derive the VLB, we first prove that for any function of $G_{0:n}$, i.e., $f(G_{0:n})$, its expectation over the diffused graph sequence is equal to the expectation over the node diffusion orderings. That is:

$$\mathbb{E}_{q_\phi(G_{1:n}|G_0)} f(G_{0:n}) = \mathbb{E}_{q_\phi(\sigma_{1:n}|G_0)} f(G_{0:n}). \tag{10}$$

*Proof.*

$$\mathbb{E}_{q_\phi(G_{1:n}|G_0)} f(G_{0:n}) = \sum_{G_{1:n}} f(G_{0:n}) q_\phi(G_{1:n}|G_0)$$

$$= \sum_{G_{1:n}} \frac{1}{|\Pi(G_{1:n})|} \sum_{\sigma_{1:n} \in \Pi(G_{1:n})} f(G_{0:n}) q_\phi(G_{1:n}|G_0)$$

$$\overset{Eq.\ 9}{=} \sum_{G_{1:n}} \frac{1}{|\Pi(G_{1:n})|} \sum_{\sigma_{1:n} \in \Pi(G_{1:n})} f(G_{0:n}) |\Pi(G_{1:n})| q_\phi(\sigma_{1:n}|G_0)$$

$$= \sum_{G_{1:n}} \sum_{\sigma_{1:n} \in \Pi(G_{1:n})} f(G_{0:n}) q_\phi(\sigma_{1:n}|G_0)$$

$$= \sum_{\sigma_{1:n}} f(G_{0:n}) q_\phi(\sigma_{1:n}|G_0)$$

$$= \mathbb{E}_{q_\phi(\sigma_{1:n}|G_0)} f(G_{0:n}). \tag{11}$$

The third last row is because iterating over the double loops of all the graph sequences and the corresponding node orderings is equal to iterating over all possible node orderings. □

Then, our VLB can be written as:

$$\log p_\theta(G_0) = \log \left( \int p_\theta(G_{0:n}) \frac{q_\phi(G_{1:n}|G_0)}{q_\phi(G_{1:n}|G_0)} dG_{1:n} \right)$$

$$\geq \mathbb{E}_{q_\phi(G_{1:n}|G_0)} \left( \log p_\theta(G_{1:n}) + \log p_\theta(G_0|G_{1:n}) - \log q_\phi(G_{1:n}|G_0) \right)$$

$$\overset{Eq.\ 10}{=} \mathbb{E}_{q_\phi(\sigma_{1:n}|G_0)} \left( \log p_\theta(G_{1:n}) + \log p_\theta(G_0|G_{1:n}) - \log q_\phi(G_{1:n}|G_0) \right)$$

$$\overset{Eq.\ 8}{=} \mathbb{E}_{q_\phi(\sigma_{1:n}|G_0)} \left( \log |\Pi(G_{1:n})| p_\theta(G_{1:n}, \sigma_{1:n}) + \log p_\theta(G_0|G_{1:n}) - \log q_\phi(G_{1:n}|G_0) \right)$$

$$\overset{Eq.\ 9}{=} \mathbb{E}_{q_\phi(\sigma_{1:n}|G_0)} \left( \log |\Pi(G_{1:n})| p_\theta(G_{1:n}, \sigma_{1:n}) + \log p_\theta(G_0|G_{1:n}) - \log |\Pi(G_{1:n})| q_\phi(\sigma_{1:n}|G_0) \right)$$

$$= \mathbb{E}_{q_\phi(\sigma_{1:n}|G_0)} \left( \log |\Pi(G_{1:n})| p_\theta(G_{1:n-1}|\sigma_{1:n}) p(\sigma_{1:n}|G_n) p(G_n) + \log p_\theta(G_0|G_{1:n}) \right.$$
$$\left. - \log |\Pi(G_{1:n})| q_\phi(\sigma_{1:n}|G_0) \right)$$

$$= \mathbb{E}_{q_\phi(\sigma_{1:n}|G_0)} \left( \log p_\theta(G_{1:n-1}|\sigma_{1:n}) + \log p_\theta(G_0|G_{1:n}) + \log p(\sigma_{1:n}|G_n) + \underbrace{\log p(G_n)}_{0} \right.$$
$$\left. - \log q_\phi(\sigma_{1:n}|G_0) + \log |\Pi(G_{1:n})| - \log |\Pi(G_{1:n})| \right)$$

$$= \mathbb{E}_{q_\phi(\sigma_{1:n}|G_0)} \left( \log p_\theta(G_{0:n-1}|\sigma_{1:n}, G_n) + \log p(\sigma_{1:n}|G_n) - \log q_\phi(\sigma_{1:n}|G_0) \right)$$

$$= \mathbb{E}_{q_\phi(\sigma_{1:n}|G_0)} \left( \log p_\theta(G_{0:n-1}(\sigma_{1:n})|G_n) + \log p(\sigma_{1:n}|G_n) - \log q_\phi(\sigma_{1:n}|G_0) \right)$$

$$= \mathbb{E}_{q_\phi(\sigma_{1:n}|G_0)} \left( \log p_\theta(G_{0:n-1}(\sigma_{1:n})|G_n) + \log p(\sigma_{1:n}|G_n) - \log q_\phi(\sigma_{1:n}|G_0) \right)$$

$$= \mathbb{E}_{q_\phi(\sigma_{1:n}|G_0)} \sum_{t=0}^{n-1} \log p_\theta(G_t(\sigma_{1:n})|G_{t+1}(\sigma_{1:n})) - \text{KL}(q_\phi(\sigma_{1:n}|G_0)|p(\sigma_{1:n}|G_n)). \tag{12}$$

We use $G_t(\sigma_{1:n})$ to denote the masked graph at time step $t$ under the node ordering $\sigma_{1:n}$. In the fifth last row, we have $\log p(G_n) = 0$ because $G_n$ is a deterministic graph wherein all the nodes are masked. For brevity, we directly use $p_\theta(G_t|G_{t+1})$ to represent $p_\theta(G_t(\sigma_{1:n})|G_{t+1}(\sigma_{1:n}))$ in Eq. 3 since $\sigma_{1:n}$ already appears in the expectation over $q_\phi(\sigma_{1:n}|G_0)$.

## A.4 TRAINING ALGORITHM OF GRAPHARM

We use a reinforcement learning (RL) procedure by sampling multiple diffusion trajectories, thereby enabling training both the diffusion ordering network $q_\phi(\sigma|G_0)$ and the denoising network $p_\theta(G_t|G_{t+1})$ using gradient descent.

Specifically, at each training iteration, we explore the diffusion ordering network by creating $M$ diffusion trajectories for each training graph $G_0^{(i)}$. Each trajectory is a sequence of graphs

$\{G_t^{(i,m)}\}_{1 \le t \le n}$ where the node decay ordering is sampled from $q_\phi(\sigma|G_0)$. The denoising network $p_\theta(G_t|G_{t+1})$ is then trained to minimize the negative VLB using stochastic gradient descent (SGD):

$$\theta_j \leftarrow \theta_{j-1} - \frac{\eta_1}{M} \nabla \sum_{i \in \mathcal{B}_{\text{train}}} \sum_m \sum_t \log p_\theta(G_t^{(i,m)}|G_{t+1}^{(i,m)}), \tag{13}$$

where $\mathcal{B}_{\text{train}}$ is the a minibatch sampled from the training data.

To evaluate the current diffusion ordering network, we create $M$ trajectories for each validation graph and compute the negative VLB of the denoising network to obtain the corresponding rewards $R_t^m = -\sum_{i \in \mathcal{B}_{\text{val}}} \sum_t \log p_\theta(G_t^{(i,m)}|G_{t+1}^{(i,m)})$. Then, the diffusion ordering network can be updated with common RL optimization methods, *e.g.*, the REINFORCE algorithm (Williams, 1992):

$$\phi_j \leftarrow \phi_{j-1} - \frac{\eta_2}{M} \sum_m R_t^m \nabla \log q_\phi(\sigma|G_0). \tag{14}$$

The detailed training procedure is summarized in Algorithm 1.

---

**Algorithm 1** Training procedure of GRAPHARM

---

**Require:** Diffusion ordering network $q_\phi(\sigma|G_0)$, Denoising network $p_\theta(G_t|G_{t+1})$
1: **for** # training iterations **do**
2:     Sample a minibatch $\mathcal{B}$ from the training set
3:     **for** each $i \in \mathcal{B}_{\text{train}}$ **do**
4:         **for** each $m \in [1, M]$ **do**
5:             $\sigma_1, \cdots, \sigma_t \sim q_\phi(\sigma|G_0)$
6:             Obtain corresponding diffusion trajectories $\{G_t^{(i,m)}\}_{1 \le t \le n}$
7:         **end for**
8:         $\theta_j \leftarrow \theta_{j-1} - \frac{\eta_1}{M} \nabla \sum_{i \in \mathcal{D}_{\text{train}}} \sum_m \sum_t \log p_\theta(G_t^{(i,m)}|G_{t+1}^{(i,m)})$
9:     **end for**
10:     Sample a minibatch from the validation set
11:     **for** each $i \in \mathcal{B}_{\text{val}}$ **do**
12:         **for** each $m \in [1, M]$ **do**
13:             $\sigma_1, \cdots, \sigma_t \sim q_\phi(\sigma|G_0)$
14:             Obtain corresponding diffusion trajectories $\{G_t^{(i,m)}\}_{1 \le t \le n}$
15:         **end for**
16:         Compute the reward $R_t^m = -\sum_{i \in \mathcal{D}_{\text{val}}} \sum_t \log p_\theta(G_t^{(i,m)}|G_{t+1}^{(i,m)})$
17:         $\phi_j \leftarrow \phi_{j-1} - \frac{\eta_2}{M} \sum_m R_t^m \nabla \log q_\phi(\sigma|G_0).$
18:     **end for**
19: **end for**

---

### A.5 EXTENDED RELATED WORK

**Graph Generation.** *One-shot graph generative models* generate all edges between nodes at once. Models based on VAEs (Simonovsky & Komodakis, 2018; Liu et al., 2018; Ma et al., 2018) and GANs (Cao & Kipf, 2022; Maziarka et al., 2020) generate all edges independently from latent embeddings. This independence assumption can hurt the quality of the generated graphs. Normalizing flow models (Zang & Wang, 2020; Madhawa et al., 2019) are restricted to invertible model architectures for building a normalized probability. Another drawback shared by all one-shot generation models is that they cannot incorporate constraints during generation due to the one-shot process. The other class is *autoregressive graph generative models*, which generate a graph by sequentially adding nodes and edges. Autoregressive generation can be achieved using recurrent networks (Li et al., 2018; You et al., 2018b; Popova et al., 2019), VAEs (Liu et al., 2018; Jin et al., 2018; 2020), normalizing flows (Shi et al., 2020; Luo et al., 2021), and RL (You et al., 2018a). By breaking the problem into smaller parts, these methods are more apt at capturing complex structural patterns and can easily incorporate constraints during generation. However, a key drawback of them is that their training is sensitive to node ordering. Most existing works pre-define a fixed node ordering by ad-hoc such as random breadth-first search (BFS) ordering (You et al., 2018b; Shi et al., 2020; Luo et al., 2021), which cannot preserve permutation invariance and also be suboptimal for generation. Chen et al. (2021) propose to learn the node generation ordering using variational inference (VI).

**Diffusion and Score-Based Generation.** Diffusion models have emerged as a new family of powerful deep generative models. Denoising diffusion probabilistic modeling (DDPM) (Sohl-Dickstein et al., 2015; Ho et al., 2020) perturbs the data distribution into a Gaussian base distribution through an forward Markov noising process, and then learns to recover data distribution via the reverse transition of the Markov chain. Closely related to DDPM is score-based generation (Song & Ermon, 2019), which perturbs data with gradually increasing noise, and then learns to reverse the perturbation via score matching. Song et al. (2021) generalize diffusion models to continuous-time diffusion using forward and backward SDEs. Diffusion and score-based models has been successfully developed for problems including image synthesis (Dhariwal & Nichol, 2021), text-to-image synthesis (Ramesh et al., 2022; Saharia et al., 2022), and molecular conformation modeling (Xu et al., 2021; Hoogeboom et al., 2022b).

A few works have proposed discrete diffusion models that operate directly in discrete state spaces through multinomial diffusion that uses categorical noise (Hoogeboom et al., 2021), discrete absorbing diffusion (Austin et al., 2021), or autoregressive diffusion (Hoogeboom et al., 2022a). Our GRAPHARM model extends autoregressive diffusion in two ways: (1) It defines autoregressive absorbing diffusion processes on graphs; (2) It learns an optimal absorbing node ordering via the permutation invariance property in graph generation. Few works have studied diffusion-based graph generation. To the best of our knowledge, Niu et al. (2020); Jo et al. (2022) are the only two works in this line. Niu et al. (2020) perturb the adjacency matrices with gradually increasing noise levels, and Jo et al. (2022) model diffusion on both the adjacency matrices and node features using continuous-time SDE-based diffusion. Different from these two works, our model is the first *autoregressive* diffusion model for graph generation, which defines diffusion directly in the discrete graph space.

## A.6 IMPLEMENTATION DETAILS

**Model optimization**: We use ADAM with $\beta_1 = 0.9$ and $\beta = 0.999$ as the optimizer. The learning rate is set for $10^{-4}$ and $5 \times 10^{-4}$ for the denoising network and diffusion ordering network respectively on all the datasets. We perform model selection based on the average MMD of the three metrics on the validation set.

**Network architecture**: For fair comparison, our denoising network uses the same graph attention network architecture as the baseline GRAN which has 7 layers and hidden dimensions 128; the diffusion ordering network uses the same graph attention network architecture as the baseline OA which is a vallina GAT (Veličković et al., 2018) that has 3 layers with 6 attention heads and residual connections. The hidden dimension is set to 32. As OM is compatible with any existing autoregressive methods, we use the strongest autoregressive baseline GRAN as its backbone.

**Hyper-parameters**: We set the number of trajectories $M$ as 4 for all the datasets. Both GRAPHARM and GRAN use 20 as the number of Bernoulli mixtures. For GRAN, we use block size 1 and stride 1 to achieve the best generation performance. For GDSS, we choose the best signal-to-noise ratio (SNR) from $\{0.05, 0.1, 0.15, 0.2\}$ and scale coefficient from $\{0.1, 0.2, 0.3, 0.4, 0.5, 0.6, 0.7, 0.8, 0.9, 1.0\}$ based on the average MMD of degree, clustering coefficient and orbit as in Jo et al. (2022). For EDP-GNN, we use 6 noise levels $\{\sigma_i\}_{i=1}^L = [1.6, 0.8, 0.6, 0.4, 0.2, 0.1]$ as suggested in the original work (Niu et al., 2020). For OM, we use 4 as the sample size for variational inference as suggested in the original work (Chen et al., 2021)

## A.7 EXPERIMENT SETUP FOR CONSTRAINED GRAPH GENERATION

We use the Caveman dataset for the constrained generation experiment. We set the constraint as that the maximum node degree is no larger than 6. To generate graphs under this constraint with GRAPHARM, we add a degree checking into the generation process. Specifically, when generating a new node and its connecting edges, we first check whether the constraint will be violated on previous nodes as new edges are added to them. If so, the corresponding edges will be dropped; otherwise we proceed to check the constraint on the new generated node and randomly remove the extra edges that exceed the limit.

## A.8 Dataset Statistics

We evaluate the performance of GRAPHARM on six diverse graph generation benchmarks, covering both synthetic and real-world graphs from different domains: (1) Community-small (You et al., 2018b), containing 100 graphs randomly generated community graphs with $12 \leq |V| \leq 20$; (2) Caveman (You, 2018), containing 200 caveman graphs synthetically generated with $5 \leq |V| \leq 10$; (3) Cora, containing 200 sub-graphs with $9 \leq |V| \leq 87$, extracted from Cora network (Sen et al., 2008) using random walk; (4) Breast, including 100 chemical graphs with $12 \leq |V| \leq 18$, sampled from Gonzalez-Malerva et al. (2011); (5) Enzymes, including 563 protein graphs with $10 \leq |V| \leq 125$ from BRENDA database (Schomburg et al., 2004); (6) Ego-small, containing 200 small sub-graphs with $4 \leq |V| \leq 18$ sampled from Citeseer Network Dataset (Sen et al., 2008). For each dataset, we use $80\%$ of the graphs as training set and the rest $20\%$ as test sets. Following (Liao et al., 2019), we randomly select $25\%$ from the training data as the validation set.

## A.9 Approximation Error of the KL-divergence Term in Eq. 3.

To evaluate the approximation error, we compute the KL-divergence between the diffusion ordering and the generation ordering at each time step and then sum together, i.e., Approximation error=$\sum_t \text{KL}\left(q(\sigma_t|G_0, \sigma_{(<t)})|p(\sigma_t|G_t)\right)$. Specifically, at each time step $t$, we use GRAPHARM to first generate a graph $G_0$ and record the corresponding node generation ordering $\sigma$. Since we do not explicitly model distribution of the node index during the generation procedure, we determine the index of the generated node in $G_0$ by its connectivity to the unmasked nodes in $G_t$. Finally, we forward the generation network multiple times to obtain an empirical distribution for $p(\sigma_t|G_t)$. Table 5 provides the approximation error versus the training iterations. As we can see, with the training progresses, the approximation error indeed becomes smaller and approaches to zero.

| Training iterations | 0 | 100 | 200 | 500 | 2000 | 3000 |
|---|---|---|---|---|---|---|
| Appriximation error | 0.343 | 0.262 | 0.131 | 0.093 | 0.061 | 0.049 |

Table 5: Approximation error between the generation ordering and the diffusion ordering on the community-small dataset.

