# OpenReview forum: "Autoregressive Diffusion Model for Graph Generation"
_ICLR.cc/2023/Conference — Submitted to ICLR 2023_

### Official Review · Reviewer_KoQP · 2022-10-21

**Confidence:** 4
**Correctness:** 4
**Technical Novelty And Significance:** 2
**Empirical Novelty And Significance:** 3
**Recommendation:** 6

**Clarity, Quality, Novelty And Reproducibility:**

Clarity and Quality: The presentation of this paper is good.

Novelty: An autoregressive diffusion model has been proposed for the graph generation, where a proper ordering can be learned via training.

Reproducibility: The authors have provided details about the training parameters and network architectures in the appendix. These would help to reproduce the experimental results.

**Strength And Weaknesses:**

Strength:

A new node-absorbing diffusion process has been introduced, and a data-dependent ordering can be learned via the optimization. The proposed method has been compared with several SOTA methods and shown its competitive results including the generation time.



######################

Weaknesses:

. Section 2: The authors mentioned that the absorbing diffusion is the most promising generation method. Can you add some explanation on that?

. Section 3.1: The diffusion ordering network produces the probability of the node at time t via equation (1). Can you explain why such an ordering would reflect the topology/regularities of the graph?

. Section 3.3: The proposed training objective has ignored the KL-divergence term in equation (3). Can you evaluate such approximation error, ie. calculate the actual KL-divergence and check whether it indeed approaches zero?

Experiments:

. Table 1: The performance of the proposed GRAPHARM on Cora is not competitive. Can you explain that?

. Effect of Diffusion Ordering: Can you illustrate the proposed ordering visually to give a sense that it does reflect the topology/regularities of the graph when compared to the random ordering?


**Summary Of The Paper:**

The authors proposed an autoregressive diffusion model for the graph generation. A node-absorbing diffusion process was introduced. For the forward diffusion a diffusion ordering network was suggested, and for the reverse diffusion a denoising network was designed. These two networks can be trained jointly with a simplified log-likelihood loss. Experiments have shown the competitive results of the proposed method when compared to several existing methods.

**Summary Of The Review:**

The authors proposed an autoregressive diffusion model for the graph generation. The analysis is solid and the proposed method has been compared with several SOTA methods and showed its competitive performance. The paper can be further improved by adding more explanations and discussions (see the comments in Weaknesses above).

---

> ### Author Response · Authors · 2022-11-19
> **Reply to Reviewer KoQP's Comments**
>
> **Q1. Section 2: The authors mentioned that the absorbing diffusion is the most promising generation method. Can you add some explanation on that?**
>
> The absorbing diffusion is the most promising generation method because (1) It achieves the best performance on both character-level and token-level text generation compared with other discrete diffusion models [1]. (2) The absorbing diffusion has already been widely used in natural language processing. [1] shows that existing masked language models, e.g., BERT, are equivalent to absorbing diffusion.
>
>
>
> **Q2. Section 3.1: The diffusion ordering network produces the probability of the node at time t via equation (1). Can you explain why such an ordering would reflect the topology/regularities of the graph?**
>
> The diffusion ordering network learns an optimal node ordering by optimizing the variational lower bound in our training objective. The optimal node ordering can reflect the graph topology/regularity better than a random ordering. We quantitatively evaluate the generation ordering of GRAPHARM on the community-small dataset and compare it with OA-ARM which uses a random generation ordering. We first generate 100 graphs using both methods and record the node generation orderings. Then, we use the graph clustering method to partition the nodes into two clusters for each generated graph. Finally, we count the nodes that cross different clusters during the generation procedure. As we can see from Table 1, the average number of nodes that cross different clusters of our method is much smaller than OA-ARM. This demonstrates that our method tends first to generate one cluster and then add another cluster while OA-ARM just randomly generates the graph. Therefore, the generation ordering of GRAPHARM can reflect the graph topology/regularity better than OA-ARM.
>
> |Method | Average Counts |
> |----|---|
> | GraphARM| 2.6|
> |OA-ARM|6.9|
>
> Table 1. Average number of nodes that cross different clusters on the community-small dataset.
>
>
>
> **Q3. Section 3.3: The proposed training objective has ignored the KL-divergence term in equation (3). Can you evaluate such approximation error, ie. calculate the actual KL-divergence and check whether it indeed approaches zero?**
>
> Thanks for the nice suggestion. We have added new results to evaluate the KL approximation error. To evaluate the approximation error, we compute the KL-divergence between the diffusion ordering and the generation ordering at each time step and then sum together, i.e., $\text{Approximation error=}\sum_{t}\text{KL}\left(q(\sigma_t|G_0,\sigma_{(<t)})|p(\sigma_t|G_{t})\right)$. Specifically, at each time step $t$, we use GRAPHARM to first generate a graph $G_0$ and record the corresponding node generation ordering $\sigma$. Since we do not explicitly model distribution of the node index during the generation procedure, we determine the index of the generated node in $G_0$ by its connectivity to the unmasked nodes in $G_{t}$. Finally, we forward the generation network multiple times to obtain an empirical distribution for $p(\sigma_t|G_{t})$. Table 2 provides the approximation error versus the training iterations. As we can see, with the training progresses, the approximation error indeed becomes smaller and approaches to zero.
>
> |Training iterations| 0|100 |200 |500 |2000 |3000 |
> |----|---|---|---|---|---|---|
> | Approximation error|0.343|0.262 | 0.131| 0.093 |0.061 |0.049 |
>
> Table 2. Approximation error between the generation ordering and the diffusion ordering on the community-small dataset.
>
>
> **Q4. Table 1: The performance of the proposed GRAPHARM on Cora is not competitive. Can you explain that?**
>
> Following [3], the Cora dataset is constructed by sampling 200 subgraphs from the Cora network [2] using random walks. Due to the random walk subsampling process, the Cora dataset has less regularity in terms of graph structural patters. That is likely why our method has less advantage on this dataset.
>
>
>
> **Q5. Effect of Diffusion Ordering: Can you illustrate the proposed ordering visually to give a sense that it does reflect the topology/regularities of the graph when compared to the random ordering?**
>
>
> We did visualize the generation ordering in Figure 3 (in appendix A.1). As we can see, GRAPHARM first generates one community and then adds another one, which shows that GRAPHARM captures graph structural topology for generation. In contrast, OA-ARM generates the graph with a random order.
>
>  ### References
>
> [1] Jacob Austin, et al, Structured Denoising Diffusion Models in Discrete State-Spaces, *NeurIPS 2021*.
>
> [2] Prithviraj Sen et al, Collective classification in network data, *AI magazine, 2008*.
>
> [3] Chen, Xiaohui, et al, Order Matters: Probabilistic Modeling of Node Sequence for Graph Generation, *ICML 2021*.

---

> > ### Comment · Reviewer_KoQP · 2022-11-23
> > **Thanks for your response.**
> >
> > Thanks for the response. The authors have generally addressed the reviewer's comments. The original score (6) has already been left some room for the improvement, so won't be further increased.

---

### Official Review · Reviewer_dB1j · 2022-10-25

**Confidence:** 5
**Correctness:** 1
**Technical Novelty And Significance:** 2
**Empirical Novelty And Significance:** 3
**Recommendation:** 1

**Clarity, Quality, Novelty And Reproducibility:**


The description of the diffusion process can be clearer. For example, the definition of "Absorbing Node State" seems to indicate that a node enters the absorbing state after/because it is masked and connected to other nodes with masked edges. But actually, the diffusion process puts the node to an "absorbing node state" -- as a consequence, it is masked and connected to other nodes with masked edges.

[After discussion], the notation system in this submission is not well defined. It should follow previous work and use adjacency matrices to derive probabilities.

**Strength And Weaknesses:**

Strength:

+ the new model has a formulation of autoregressive diffusion model.
+ the model is very efficient
+ the model shows good generation performances in a list of tasks.

Weaknesses

- Though the formulation is motivated differently, it has a large overlap with (Chen et al., 2021). The diffusion ordering network is roughly the same as the variational distribution of node orders in (Chen et al., 2021). The variational lower bound is also very similar to (Chen et al. 2021). The difference is actually because this work neglects the difference between graph sequences and node orders: a graph sequence is not equivalent to a node ordering, which is analyzed by (Chen et al., 2021).
- The generative procedure is also similar to the standard autoregressive generative procedure: every time adding a new node and the edges connecting to this new node. I don't see the benefit of describing this procedure using the diffusion process.
- There are several flaws in the analysis. The first one is the difference mentioned above, then the variational lower bound is not exactly correct. The second one is the complexity of generation: the complexity of this model should still be O(n^2) (the analysis below equation 2). Anyhow the model needs to make a decision for each node pair. The running time is saved by reducing the number of calls of neural networks (from O(n^2) to O(n)).

[After a careful study of the submission and a discussion with authors], I find a critical derivation error in the submission. In particular, the submission bases its derivation on equation 8, which is from (Chen et al., 2021), but the term $p(G_{1:n}, \sigma_{1:n})$ in this submission is different from that in the previous work, so the derivation is incorrect.

**Summary Of The Paper:**

The paper proposes a new graph generative model based on an autoregressive diffusion model. It shows good performances in a list of tasks.

**Summary Of The Review:**

The work devises a new graph generative model from the perspective of autoregressive diffusion models. However, it does not seem to add much to the original formulation. The model design is a contribution: it is efficient and shows good performance in several tasks.

[After discussion:] the notation system of the submission is not well defined, and the derivation is problematic.

---

> ### Author Response · Authors · 2022-11-19
> **Reply to Reviewer dB1j's Comments, Part I**
>
>
>
> **Q1. The variational lower bound is not exactly correct because this work neglects the difference between graph sequences and node orders: a graph sequence is not equivalent to a node ordering, which is analyzed by (Chen et al., 2021).**
>
>
>
> Our variational lower bound (VLB) in Eq.3 is **correct**. This is because for autoregressive diffusion, the latent variable is the graph sequence defined by the diffusion steps, which can eliminate the need of modeling graph automorphism in the corresponding VLB. In contrast, Chen et. al. [1] treat the node ordering as the latent variable and thus must account for graph automorphism in their VLB. Let us illustrate in more details below.
>
> Following [1], let us define $\mathit{\Pi}(G_{1:n})$ as the set of all node orderings that give the same graph sequence $G_{1:n}$, i.e., $\mathit{\Pi}(G_{1:n})=\\{\sigma_{1:n}: G_t(\sigma_{1:t})=G_t, \forall t=1,\cdots,n \\}$ where $G_t(\sigma_{1:t})$ is the masked graph at time step $t$ under the node ordering $\sigma_{1:t}$. Following [1], the conditional $p(\sigma_{1:n}|G_{1:n})$ is a uniform distribution, i.e., $p(\sigma_{1:n}|G_{1:n})=\frac{1}{|\mathit{\Pi}(G_{1:n})|}, \forall \sigma_{1:n}\in\mathit{\Pi}(G_{1:n})$.
> Then, we have the following two equations:
>
> (1) $p_{\theta}(G_{1:n},\sigma_{1:n})=\frac{1}{|\mathit{\Pi}(G_{1:n})|}p_{\theta}(G_{1:n}), \quad (r.1)$
>
> (2) $q_{\phi}(G_{1:n}|G_0)=|\mathit{\Pi}(G_{1:n})|q_{\phi}(\sigma_{1:n}|G_0) \quad (r.2)$.
>
>
>
> To derive the VLB, we first prove that for any function of $G_{0:n}$, i.e., $f(G_{0:n})$, its expectation over the graph sequence is equal to the expectation over the node ordering. That is:
>
> $E_{q(G_{1:n}|G_0)}f(G_{0:n})= E_{q(\sigma_{1:n}|G_0)}f(G_{0:n}). \quad (r.3)$
>
> Proof:
>
> \begin{align}
> E_{q_{\phi}(G_{1:n}|G_0)}f(G_{0:n}) =& \sum_{G_{1:n}} f(G_{0:n})q_{\phi}(G_{1:n}|G_0)\nonumber \\\\
> =&\sum_{G_{1:n}}\frac{1}{|\mathit{\Pi}(G_{1:n})|} \sum_{\sigma_{1:n}\in \mathit{\Pi}(G_{1:n})} f(G_{0:n})q_{\phi}(G_{1:n}|G_0)\nonumber \\\\
> \overset{Eq.(r.2)}{=}&\sum_{G_{1:n}}\frac{1}{|\mathit{\Pi}(G_{1:n})|}\sum_{\sigma_{1:n}\in \mathit{\Pi}(G_{1:n})}f(G_{0:n})|\mathit{\Pi}(G_{1:n})|q_{\phi}(\sigma_{1:n}|G_0)\nonumber\\\\
> =&\sum_{G_{1:n}}\sum_{\sigma_{1:n}\in \mathit{\Pi}(G_{1:n})}f(G_{0:n})q_{\phi}(\sigma_{1:n}|G_0)\nonumber\\\\
> =&\sum_{\sigma_{1:n}} f(G_{0:n})q_{\phi}(\sigma_{1:n}|G_0)\nonumber\\\\
> =&E_{q_{\phi}(\sigma_{1:n}|G_0)}f(G_{0:n}).
> \end{align}
> The third last row is because iterating over the double loops of all possible graph sequences and the corresponding node orderings is equal to iterating over all possible node orderings.

---

> > ### Author Response · Authors · 2022-11-19
> > **Reply to Reviewer dB1j's Comments, Part II**
> >
> > Then, our VLB can be written as:
> > \begin{align}
> >   \log{p_{\theta}(G_{0})}  =&  \log{\left(\int p_{\theta}(G_{0:n})\frac{q_{\phi}(G_{1:n}|G_0)}{q_{\phi}(G_{1:n}|G_0)}dG_{1:n}\right)} \nonumber\\\\
> >                           \geq& E_{q_{\phi}(G_{1:n}|G_0)}\left( \log{p_{\theta}(G_{1:n})} + \log{p_{\theta}(G_0|G_{1:n})} - \log{q_{\phi}(G_{1:n}|G_{0})} \right) \nonumber\\\\
> >                           \overset{Eq.(r.3)}{=}&  E_{q_{\phi}(\sigma_{1:n}|G_0)}\left( \log{p_{\theta}(G_{1:n})} + \log{p_{\theta}(G_0|G_{1:n})} - \log{q_{\phi}(G_{1:n}|G_{0})} \right) \nonumber \\\\
> >                           \overset{Eq.(r.1)}{=}& E_{q_{\phi}(\sigma_{1:n}|G_0)}\left( \log{|\mathit{\Pi}(G_{1:n})|p_{\theta}(G_{1:n},\sigma_{1:n})} + \log{p_{\theta}(G_0|G_{1:n})} - \log{q_{\phi}(G_{1:n}|G_{0})} \right) \nonumber
> >                           \\\\
> >                                          \overset{Eq.(r.2)}{=}& E_{q_{\phi}(\sigma_{1:n}|G_0)}\left( \log{|\mathit{\Pi}(G_{1:n})|p_{\theta}(G_{1:n},\sigma_{1:n})} + \log{p_{\theta}(G_0|G_{1:n})} - \log{|\mathit{\Pi}(G_{1:n})|q_{\phi}(\sigma_{1:n}|G_{0})} \right) \nonumber
> >                           \\\\
> >                           =& E_{q_{\phi}(\sigma_{1:n}|G_0)}( \log{|\mathit{\Pi}(G_{1:n})|p_{\theta}(G_{1:n-1}|\sigma_{1:n})p(\sigma_{1:n}|G_n)p(G_n)} + \log{p_{\theta}(G_0|G_{1:n})}  \nonumber\\\\
> >                           & - \log{|\mathit{\Pi}(G_{1:n})|q_{\phi}(\sigma_{1:n}|G_{0})} ) \nonumber \\\\
> >                               =& E_{q_{\phi}(\sigma_{1:n}|G_0)} ( \log{p_{\theta}(G_{1:n-1}|\sigma_{1:n})} + \log{p_{\theta}(G_0|G_{1:n})} +\log{p(\sigma_{1:n}|G_n)} +\log{p(G_n)}\nonumber \\\\
> >                                                     & - \log{q_{
> >                               \phi}(\sigma_{1:n}|G_{0})}
> >                             + \log{|\mathit{\Pi}(G_{1:n})|}-\log{|\mathit{\Pi}(G_{1:n})|} ) \nonumber \\\\
> >  \overset{\log{p(G_n)=0}}{=}& E_{q_{\phi}(\sigma_{1:n}|G_0)} ( \log{p_{\theta}(G_{1:n-1}|\sigma_{1:n})} + \log{p_{\theta}(G_0|G_{1:n})} +\log{p(\sigma_{1:n}|G_n)} - \log{q_{
> >                               \phi}(\sigma_{1:n}|G_{0})} \nonumber \\\\\
> >                                                     =& E_{q_{\phi}(\sigma_{1:n}|G_0)}( \log{p_{\theta}(G_{0:n-1}|\sigma_{1:n}, G_n)} +\log{p(\sigma_{1:n}|G_n)} - \log{q_{\phi}(\sigma_{1:n}|G_{0})}
> >             )\nonumber \\\\
> >                                                                =& E_{q_{\phi}(\sigma_{1:n}|G_0)}\left( \log{p_{\theta}(G_{0:n-1}(\sigma_{1:n})|G_n)} +\log{p(\sigma_{1:n}|G_n)}- \log{q_{\phi}(\sigma_{1:n}|G_{0})}
> >                             \right)\nonumber \\\\
> >                                                         =& E_{q_{\phi}(\sigma_{1:n}|G_0)}\left( \log{p_{\theta}(G_{0:n-1}(\sigma_{1:n})|G_n)}+ \log{p(\sigma_{1:n}|G_n)}- \log{q_{\phi}(\sigma_{1:n}|G_{0})}
> >                             \right)\nonumber \\\\
> >                           =& E_{q_{\phi}(\sigma_{1:n}|G_0)}
> >                             \sum_{t=0}^{n-1}\log{p_{\theta}(G_t(\sigma_{1:n})|G_{t+1}(\sigma_{1:n}))}-\text{KL}(q_{\phi}(\sigma_{1:n}|G_0)|p(\sigma_{1:n}|G_n)).
> > \end{align}
> > We use $G_t(\sigma_{1:n})$ to denote the masked graph at time step $t$ under the node ordering $\sigma_{1:n}$. In the fifth last row, we have $\log p(G_n)=0$ because $G_n$ is a deterministic graph wherein all the nodes are masked. For brevity, we directly use $p_{\theta}(G_t|G_{t+1})$ to represent $p_{\theta}(G_t(\sigma_{1:n})|G_{t+1}(\sigma_{1:n}))$ in Eq. 3 since $\sigma_{1:n}$ already appears in the expectation over $q_{\phi}(\sigma_{1:n}|G_0)$.
> >
> >
> > We have also added this detailed derivation of the VLB in Appendix. A.3
> >
> >
> > In contrast, OM treats the node ordering as the latent variable and its VLB follows:
> >
> > \begin{align}
> > \log p_{\theta}(G)=& \log \sum_{\sigma_{1:n}} p_{\theta}(G_{1:n},\sigma_{1:n}) \\\\
> > =& \log \sum_{\sigma_{1:n}} \frac{p_{\theta}(G_{1:n},\sigma_{1:n})q(\sigma_{1:n}|G)}{q(\sigma_{1:n}|G)} \\\\ \ge& E_{q(\sigma_{1:n}|G)}[\log p_{\theta}(G_{1:n},\sigma_{1:n})-\log q(\sigma_{1:n}|G)] \\\\
> > =&E_{q(\sigma_{1:n}|G_{0})}[\log \frac{1}{\Pi[G_{1:n}]}p_{\theta}(G_{1:n})-\log q(\sigma_{1:n}|G)]
> > \end{align}
> >
> > As we can see, OM must compute $|\mathit{\Pi}(G_{1:n})|$ because it needs to model the joint probability $p(G,\sigma)$.

---

> > > ### Author Response · Authors · 2022-11-19
> > > **Reply to Reviewer dB1j's Comments, Part III**
> > >
> > > **Q2. Though the formulation is motivated differently, it has a large overlap with (Chen et al., 2021). The diffusion ordering network is roughly the same as the variational distribution of node orders in (Chen et al., 2021). The variational lower bound is also very similar to (Chen et al. 2021). The difference is actually because this work neglects the difference between graph sequences and node orders: a graph sequence is not equivalent to a node ordering, which is analyzed by (Chen et al., 2021).**
> > >
> > > Our variational lower bound (VLB) in Eq.3 is different from OM [1] in three aspects:
> > >
> > > (1) The VLB in OM includes the entropy of the posterior node ordering while ours does not. This is because OM treats the node ordering as the latent variable and infers its posterior distribution which is similar to variational autoencoder (VAE). However, existing works have shown that this entropy term in VAE can cause posterior collapse [2,3]. While our method is based on autoregressive diffusion and has a simpler training objective. We believe this is a key reason why our method consistently outperforms OM empirically.
> > >
> > > (2) The VLB in OM needs to compute $|\mathit{\Pi}[G_{1:n}]|$ which requires some complicated approximation algorithms such as color refinement. Our method avoids this computation because of the diffusion process framework as we have shown in Q1.
> > >
> > > (3) In our VLB, we have a KL-divergence term between the generation ordering and the diffusion ordering. However, since graph generation is node permutation invariant, the first term in Eq. 3, i.e., $E_{q(\sigma_{1},\cdots,\sigma_{n}|G_0)}
> > >                             \sum_{t}\log{p_{\theta}(G_t|G_{t+1})}$, encourages the model to generate the nodes in the exact reverse ordering of the diffusion process. This pushes the generation ordering close to the diffusion ordering, and the KL-divergence term will converge to 0. Therefore, we can ignore this term during training.
> > >
> > > As we illustrate in Q1, the difference of our VLB is because we treat the graph sequences in the diffusion process as the latent variables while OM treats the node ordering as the latent variable.
> > >
> > > **Q3. The generative procedure is also similar to the standard autoregressive generative procedure: every time adding a new node and the edges connecting to this new node. I don't see the benefit of describing this procedure using the diffusion process.**
> > >
> > >
> > >
> > > As we illustrate in Q1 and Q2, the VLB derived from our diffusion process is much simpler than that derived from the standard autoregressive model as in [1]. This is because the diffusion process treats the graph sequences as the latent variable while the [1] treat the node ordering as the latent variable. The VLB derived from our diffusion process does not need to compute $|\mathit{\Pi}(G_{1:n})|$ and avoid the posterior collapse issue. Our experimental results also verify that our proposed GRAPHARM can outperform existing autoregressive methods consistently including OM [1].
> > >
> > > **Q4. The complexity of this model should still be O(n^2) (the analysis below equation 2). Anyhow the model needs to make a decision for each node pair. The running time is saved by reducing the number of calls of neural networks (from O(n^2) to O(n)).**
> > >
> > > We have updated the description of the complexity in the manuscript to make it more precise. We meant that our model reduces the sequential generation steps from $\mathcal{O}(n^2)$ to $\mathcal{O}(n)$ to improve generation efficiency.
> > >
> > > **Q5. The description of the diffusion process can be clearer. For example, the definition of "Absorbing Node State" seems to indicate that a node enters the absorbing state after/because it is masked and connected to other nodes with masked edges. But actually, the diffusion process puts the node to an "absorbing node state" -- as a consequence, it is masked and connected to other nodes with masked edges.**
> > >
> > > Thanks for the nice suggestion. We have modified the description of the diffusion process in the updated manuscript.
> > >
> > > ### References
> > >
> > > [1] Chen, Xiaohui, et al, Order Matters: Probabilistic Modeling of Node Sequence for Graph Generation, *ICML 2021*.
> > >
> > > [2] Lucas, James, et al, Understanding Posterior Collapse in Generative Latent Variable Models. *ICLR 2019 Workshop*.
> > >
> > > [3] Lucas, James, et al, Don't Blame the ELBO! A Linear VAE Perspective on Posterior Collapse. *NeurIPS 2019*.

---

> > > ### Comment · Reviewer_dB1j · 2022-11-21
> > > **The derivation is still unclear**
> > >
> > > In general, you won't be able to eliminate a term by using a different derivation method. Your method and OM essentially derivate the same variational lower bound, so the term  $\log |\Pi(G_{1:n})|$ needs to be somewhere in your result.
> > >
> > > The statement "OM must compute  $\log |\Pi(G_{1:n})|$  because it needs to model the joint probability $p(G, \sigma)$" is incorrect. The term is obtained because OM models $p(\mathbf{A})$, which is $|\Pi(G_{1:n})|$ times of $p(G, \sigma)$.
> > >
> > > I feel that your new notation $p_\theta (G_{0: n-1}(\sigma_{1:n}) |G_n ) = p(G_{0: n-1} |\sigma_{1:n}, G_n )$ is problematic. From this probability, you get $\sum_{t=0}^{n-1} p_\theta(G_t(\sigma_{1:n}) | G_{t+1}(\sigma_{1:n}))$. What does $G_t(\sigma_{1:n}) $ exactly mean?
> > >
> > > Compared with the OM paper, which works on adjacency matrices, your conditional probability $p_\theta(G_t(\sigma_{1:n}) | G_{t+1}(\sigma_{1:n}))$ seems to correspond the probability of $p(A_{1:(n-t)} | A_{1:n-t-1})$ but differ by the probability of a particular node leading the growth of the adjacency matrix from $A_{1:(n - t - 1), 1:(n - t - 1)}$ to $A_{1:(n - t), 1:(n - t)}$. Because of automorphism, the probability is one over the number of nodes that lead $A_{1:(n - t - 1), 1:(n - t - 1)}$ to $A_{1:(n - t), 1:(n - t)}$. Because of this term, you don't have $\log |\Pi(G_{1:n})|$, while OM's result has.
> > >
> > > In your implementation, your network seems to also compute $p(A_{1:(n - t), 1:(n - t)}) | A_{1:(n - t - 1), 1:(n - t - 1)})$ as other autoregressive models do, so your derivation of the lower bound does not seem to match your implementation.
> > >
> > > There is a separate issue in your derivation: $p(\sigma_{1:n} |G_n)$ is unknown. According to your model, this distribution cannot be defined: it needs to be derived by marginalizing out $G_{0:n-1}$ from $p(G_{0:n-1}, \sigma_{1:n} | G_n)$. By an argument of symmetry, it seems that the probability $p(\sigma_{1:n} |G_n)$ is  $\frac{1}{n!}$, but it needs formal proof. The submission just says it "is the distribution of the generation ordering" without a concrete calculation.
> > >
> > > I hope these derivation issues can be corrected.

---

> > > > ### Author Response · Authors · 2022-11-23
> > > > **Reply to the Derivation, Part I**
> > > >
> > > > Thanks for your response! We provide our responses as follows:
> > > >
> > > > **Q1. In general, you won't be able to eliminate a term by using a different derivation method. Your method and OM essentially derivate the same variational lower bound, so the term  needs to be somewhere in your result.**
> > > >
> > > >
> > > > We want to clarify our VLB differs from OM's more than just eliminating one term. There are three differences between the two VLBs: (1) Ours does not need to compute $|\mathit{\Pi}(G_{1:n})|$; (2) Ours does not include the entropy of node ordering; (3) Ours has an additional KL divergence term between the diffusion ordering and the generation ordering.
> > > >
> > > >
> > > > The VLBs of the two methods are different because their modeling processes are different. The key is how $p_\theta\left(G_{1: n}, \sigma_{1: n}\right)$ is treated. In OM, $p_\theta\left(G_{1: n}, \sigma_{1: n}\right)$ is factorized as: $$p_\theta\left(G_{1: n}, \sigma_{1: n}\right) = p_\theta\left(G_{1: n}\right) p_\theta\left(\sigma_{1: n} \mid G_{1: n} \right).$$
> > > >
> > > > As $p_\theta\left(\sigma_{1: n} \mid G_{1: n} \right)$ is a uniform distribution, OM inevitably needs to involve $|\mathit{\Pi}(G_{1:n})|$ in their VLB.
> > > >
> > > > In contrast, in our method  (please refer to line 6 in the derivation of VLB in our previous response), we factorize $p_\theta\left(G_{1: n}\right)$ as:
> > > >
> > > > $$ p_\theta\left(G_{1: n}, \sigma_{1: n}\right) = p_\theta\left(G_{1: n-1} \mid \sigma_{1: n}\right) p\left(\sigma_{1: n} \mid G_n\right) p\left(G_n\right).$$
> > > >
> > > > This factorization leads to the following consequences/benefits:
> > > >
> > > > - The challenge of modeling the node ordering variable is now pushed into the term $p\left(\sigma_{1: n} \mid G_n\right)$. This is a key reason that we do not have the $\left|\Pi\left(G_{1: n}\right)\right|$ term in our VLB, instead we have a KL term. Now how do we learn this conditional distribution? It turns out we do not need to! During training, the generation ordering is set to the exact reverse of the diffusion ordering and thus the KL-divergence term is equal to 0. At test time, the generator network will implicitly result in an ordering matching the diffusion ordering, and the KL-divergence term tends to approach 0 (elaborated in Q5). In other words, the node ordering network $q_\phi\left(\sigma_{1:n} \mid G_0\right)$ itself is a proxy for $p\left(\sigma_{1: n} \mid G_n\right)$. This is why we do not have the entropy term and the term $\left|\Pi\left(G_{1: n}\right)\right|$ in our VLB, and meanwhile we can also ignore the KL term but only optimizing the first term in Eq 3.
> > > >
> > > > - Still because of the factorization, our generator network $p_{\theta}(G_{0:n-1}|G_n,\sigma_{1:n})$ models the graph generation **conditioned on** a fixed node ordering. In other words, given an oracle ordering of how the nodes should be generated (which is given by the diffusion ordering network $q$), we use the generator network to model how the graph sequences are generated. In contrast, OM models $p(G_{1:n})$ with no exact node ordering information (elaborated in Q3). Our conditional generator $p_{\theta}(G_{0:n-1}|G_n,\sigma_{1:n})$ is further factorized by autoregressive generation: $\log p_{\theta}(G_{0:n-1}|G_n,\sigma_{1:n}) = \sum_{t=0}^{n-1} \log p_\theta\left(G_t\mid G_{t+1}, \sigma_{1: n}\right).$
> > > >
> > > >
> > > >
> > > > Deriving the lower bounds of the same objective from different perspectives can indeed lead to different results. This also happens in the vallina VAE and diffusion model. VAE and diffusion model are both  deriving a lower bound of  $p(\mathbf{x})$. However, their VLBs are different because their latent variables are different. Similarly, this happens between our method and OM. The modeling processes of the two are different, and their VLBs are different.
> > > >
> > > >
> > > >
> > > > **Q2. What does $G_t(\sigma_{1:n})$ exactly mean?**
> > > >
> > > > $G_{t}(\sigma_{1:n})$ means the node ordering of the graph is $\sigma_{1:n}$. We used the notation $p_\theta\left(G_t\left(\sigma_{1: n}\right) \mid G_{t+1}\left(\sigma_{1: n}\right)\right)$ to mean the autoregressive generation is conditioned on a node ordering. Namely,
> > > > $p_\theta\left(G_t\left(\sigma_{1: n}\right) \mid G_{t+1}\left(\sigma_{1: n}\right)\right) \equiv  p_\theta\left(G_t \mid G_{t+1}, \sigma_{1: n}\right)$.
> > > > We can change this notation in the final version to avoid confusion.

---

> > > > > ### Author Response · Authors · 2022-11-23
> > > > > **Reply to the Derivation, Part II**
> > > > >
> > > > >
> > > > > **Q3. Compared with the OM paper, which works on adjacency matrices, your conditional probability $p_{\theta}(G_t(\sigma_{1:n})|G_{t+1}(\sigma_{1:n}))$ seems to correspond the probability of $p(A_{1:(n-t)}|A_{1:n-t-1})$ but differ by the probability of a particular node leading the growth of the adjacency matrix from $A_{1:(n-t),1:(n-t)}$  to $A_{1:(n-t-1),1:(n-t-1)}$. Because of automorphism, the probability is one over the number of nodes that lead $A_{1:(n-t-1),1:(n-t-1)}$ to $A_{1:(n-t),1:(n-t)}$ . Because of this term, you don't have $\text{log}|\mathit{\Pi}(G_{1:n})|$, while OM's result has.**
> > > > >
> > > > > The physical meaning of $p_\theta\left(G_t\left(\sigma_{1:n}\right) \mid G_{t+1}\left(\sigma_{1: n}\right)\right) \equiv  p_\theta\left(G_t \mid G_{t+1}, \sigma_{1: n}\right)$ is different from $p(A_{1:(n-t),1:(n-t)}|A_{1:(n-t-1),1:(n-t-1)})$ and the probability of a particular node ordering leading the growth of the adjacency matrix. It corresponds to the probability of the edges between the node $v_{\sigma_t}$ with the set of unmasked nodes that have been generated so far $\\{v_{\sigma_{j}}\\}_{j=t+1}^n$, namely
> > > > >
> > > > > $p_{\theta}(G_t|G_{t+1},\sigma_{1:n})= p_{\theta}(\\{e_{\sigma_t,\sigma_j}\\}_{j=t+1}^{n}|G(t+1), \sigma(1:n))$,
> > > > >
> > > > > where we use $e_{\sigma_t,\sigma_j}$ to denote the edge between node $v_{\sigma_{t}}$ and $v_{\sigma_{j}}$.
> > > > > We use $G(t+1)$ and $\sigma(t+1)$ to denote $G_{t+1}$ and $\sigma_{t+1}$ here because the openreview does not display them in this formula properly. Note that this probability is different from the growth of the adjacency matrix, as we illustrate below.
> > > > >
> > > > > In OM, it considers the adjacency matrix $A_{1:n,1:n}$ by autoregressively modeling it as $p(A_{1:n,1:n})=\prod_tp(A_{1:(n-t),1:(n-t)})|A_{1:(n-t-1),1:(n-t-1)})$ using a neural network. However, the node ordering information is lost in $A_{1:n,1:n}$ since multiple ordering can lead to this matrix. That is why OM needs to compute automorphism to obtain $p(A_{1:n.1:n},\sigma_{1:n})=p(\sigma_{1:n}|A_{1:n,1:n})p(A_{1:n,1:n})=\frac{1}{|\mathit{\Pi}(A_{1:n,1:n})|}p(A_{1:n,1:n})$.
> > > > >
> > > > >
> > > > > In constrast, we factorize $p_{\theta}(G_{1:n},\sigma_{1:n})$ as  $p_\theta\left(G_{1: n}, \sigma_{1: n}\right) = p_\theta\left(G_{1: n-1} \mid \sigma_{1: n}\right) p\left(\sigma_{1: n} \mid G_n\right) p\left(G_n\right)$, where our generation procudure $p(G_{1:n}|\sigma_{1:n})$ is **conditioned on** a fixed node ordering $\sigma_{1:n}$. In our framework, the node ordering $\sigma_{1:n}$ is sampled in the forward diffusion process which essentially assigns a unique ID to each node of $G_0$. In the generation procedure, our starting point is the masked graph $G_n$ with $n$ nodes. Our generator network sequentially denoises the node $v_{\sigma_t}$ in $G_n$ in the reverse order of the diffusion ordering. That is, at each generation step, it computes the connectivity of node $v_{\sigma_t}$ with all unmasked nodes that have been generated so far:
> > > > > $p_{\theta}(G_t|G_{t+1},\sigma_{1:n})= p_{\theta}(\\{e_{\sigma_t,\sigma_j}\\}_{j=t+1}^{n}|G(t+1), \sigma(1:n))$.
> > > > >
> > > > > As for the generation ordering $p(\sigma_{1:n}|G_n)$ , we do not need to model it since the diffusion ordering network is a proxy (illustrate in Q1 and Q5). Therefore, by our new factorization and modeling process, we eliminate the need of computing the graph automorphism.
> > > > >
> > > > >
> > > > > **Q4. In your implementation, your network seems to also compute $p(A_{1:(n-t-1),1:(n-t-1)}|A_{1:(n-t-1),1:(n-t-1)})$ as other autoregressive models do, so your derivation of the lower bound does not seem to match your implementation.**
> > > > >
> > > > > Our derivation of the lower bound matches our implementation. At each time step $t$, our network predicts the connectivity of the node $v_{\sigma_t}$ with all the unmasked nodes that have been generated so far, i.e., $p_{\theta}(G_t|G_{t+1},\sigma_{1:n})= p_{\theta}(\\{e_{\sigma_t,\sigma_j}\\}_{j=t+1}^{n}|G(t+1), \sigma(1:n))$.
> > > > >
> > > > > The physical meaning is different from $p(A_{1:(n-t-1),1:(n-t-1)}|A_{1:(n-t-1),1:(n-t-1)})$ as we have illustrated in Q3. $A_{1:(n-t-1),1:(n-t-1)}$ is the partial adjacency matrix with no exact node ordering information. While our generation procedure starts from the masked graph $G_n$ with $n$ nodes and the generator sequentially denoises the node $v_{\sigma_t}$ in the graph in the reverse order of the diffusion ordering.

---

> > > > > > ### Author Response · Authors · 2022-11-23
> > > > > > **Reply to the Derivation, Part III**
> > > > > >
> > > > > > **Q5. There is a separate issue in your derivation:  $p(\sigma_{1:n}|G_n)$ is unknown. According to your model, this distribution cannot be defined: it needs to be derived by marginalizing out $G_{0:n-1}$ from $p(G_{0:n-1},\sigma_{1:n}|G_n)$. By an argument of symmetry, it seems that the probability $p(\sigma_{1：n}|G_n)$ is $\frac{1}{n!}$, but it needs formal proof. The submission just says "it is the distribution of the generation ordering" without a concrete calculation.**
> > > > > >
> > > > > > $p(\sigma_{1:n}|G_n)$ is the denoising/generation ordering of the generator network.  During training, the generator network denoises/generates the nodes in the graph $G_n$ using the exact reverse order of the diffusion ordering $\sigma_{1:n}$. Therefore, in training, $p(\sigma_{1:n}|G_n)$ is equal to $q(\sigma_{1:n}|G_0)$; the KL divergence term is equal to 0 and we can ignore it.
> > > > > >
> > > > > > Now the question is, at test time, how do we get $p(\sigma_{1:n}|G_n)$ as we do not have a diffusion ordering? The answer is we do not need to, because the autoregressive procedure of the generator network will implicitly result in such an ordering. During training, the training objective  $E_{q(\sigma_{1:n}|G_0)}\sum_{t}\log p(G_t|G_{t+1},\sigma_{1:n})$ encourages the generator network to denoise/generate the nodes in the reverse ordering of the diffusion process. Therefore, at test time, the generation ordering will be close to the diffusion ordering of the target graph, and the KL-divergence term tends to approach  0 (we have shown this in Appendix A.9 of the updated manuscript). Since the diffusion ordering learns an data-dependent node ordering, the matching generator network can also result in generation orderings that reflect graph topology/regularity.

---

> > > > > > > ### Comment · Reviewer_dB1j · 2022-11-26
> > > > > > > **Not answering my question**
> > > > > > >
> > > > > > > As I mentioned in my comment, your $p(\sigma_{1:n} | G_n)$ needs to be derived from your model --  you cannot say that you ``use'' the distribution of node orders from the diffusion process.
> > > > > > >
> > > > > > > Similarly, you cannot say that you use your **true posterior** in a variational inference problem (e.g. VAE) as your **variational distribution** -- you don't have the freedom to do so.

---

> > > > > > > > ### Author Response · Authors · 2022-11-28
> > > > > > > > **The generation ordering is modeled independently because of our factorization**
> > > > > > > >
> > > > > > > > We want to clarify that (1) $p(\sigma_{1:n}|G_n)$ can be modeled independently --- instead of having to be derived by marginalizing $p(G_{0:n-1},\sigma_{1:n}|G_n)$ --- because of how $p(G_{0:n-1},\sigma_{1:n}|G_n)$ is factorized in our model; (2) $q(\sigma_{1:n}|G_0)$ and $p(\sigma_{1:n}|G_n)$ model two different things in autoregressive diffusion models; their relation is not as using a variational distribution to approximate the true posterior in VAE. When training the denoising network, one thus does have the freedom to set $p(\sigma_{1:n}|G_n)$, and the VLB is maximized when and only when $p(\sigma_{1:n}|G_n) = q(\sigma_{1:n}|G_0)$.
> > > > > > > >
> > > > > > > >
> > > > > > > > First, our factorization leads to $p(G_{0:n-1},\sigma_{1:n}|G_n)=p(G_{0:n-1}|\sigma_{1:n}, G_n)p(\sigma_{1:n}|G_n)$, where $p(G_{0:n-1}|\sigma_{1:n}, G_n)$ is our generation model.
> > > > > > > > Given $G_n$, the ordering $\sigma_{1:n}$ is an independent variable, while $G_{0:n-1}$ is dependent on $\sigma_{1:n}$. Therefore, $p(\sigma_{1:n}|G_n)$ can be modeled independently --- this is akin to the freedom of parameterizing independent variables in classic graphical models.
> > > > > > > >
> > > > > > > >
> > > > > > > > Second,  in autoregressive diffusion [1], $q(\sigma_{1:n}|G_0)$ is not a variational distribution of $p(\sigma_{1:n}|G_n)$. They are modeling different things: the diffusion ordering $q(\sigma_{1:n}|G_0)$ is conditioned on the original graph $G_0$; the generation ordering $p(\sigma_{1:n}|G_n)$ is conditioned on the masked graph $G_n$. The forward diffusion process masks the graph governed by distribution $q(\sigma_{1:n}|G_0)$, while the backward generation process denoises the masked graph governed by another distribution $p(\sigma_{1:n}|G_n)$. During training, the VLB is maximized when and only when the KL-divergence term is 0, i.e., $p(\sigma_{1:n}|G_n)=q(\sigma_{1:n}|G_0)$. Therefore, when training the generation network $p(G_t|G_{t+1},\sigma_{1:n})$, we can directly set $p(\sigma_{1:n}|G_n)=q(\sigma_{1:n}|G_0)$ and ignore the KL-divergence term. A similar derivation can be found in the original autoregressive diffusion model [1] (page 21).
> > > > > > > >
> > > > > > > >
> > > > > > > > ### References
> > > > > > > >
> > > > > > > > [1] Autoregressive Diffusion Model, Emiel Hoogeboom et, al., ICLR 2022

---

> > ### Comment · Reviewer_dB1j · 2022-12-04
> > **I think I find the mistake in the derivation**
> >
> > In your derivation, one critical step is $p_\theta(G_{1:n}) = |\Pi(G_{1:n})| p_{\theta}(G_{1:n}, \sigma_{1:n})$ in equation 8. I feel this equation is wrong.
> >
> > According to your model decomposition, $p_\theta(G_{1:n}, \sigma_{1:n}) = p_{\theta}(G_{1:n-1} | \sigma_{1:n}) p(\sigma_{1:n} | G_n) p(G_{n})$. Since $ p_\theta(G_{1:n-1} | \sigma_{1:n}) $ is implemented by a GNN, which does not use node orders as the input, $ p_\theta(G_{1:n-1} | \sigma_{1:n})$ is the same for any $\sigma_{1:n}$. Then by marginalizing out $\sigma_{1:n}$, we have
> >
> > $p_\theta(G_{1:n}) = p_{\theta}(G_{1:n-1} | \sigma_{1:n})p(G_n)$
> >
> > for an arbitrary $\sigma_{1:n}$. If you decide to use an uniform distribution $p(\sigma_{1:n} | G_n) = \frac{1}{n!}$, then equation 8 should be:
> >
> > $p_\theta(G_{1:n}) = (n!)  p_{\theta}(G_{1:n}, \sigma_{1:n})$.
> >
> > With this explanation, you will have the same $|\log |\Pi(G_{1:n})|$ term as [Chen et al., 2021].
> >
> > However, there is one question not answered: why Chen et al. also use the same equation as (8)? Is their equation wrong as well? After I studied their paper, I feel that your $p(G_{1:n}, \sigma_{1:n})$ is different from theirs. If I assume your distribution $p(\sigma_{1:n} | G_n)$ is uniform, then $p(G_{1:n}, \sigma_{1:n})$  is the same for all $\sigma_{1:n}$, but their $p(G_{1:n}, \sigma_{1:n})$ does not have the behavior.
> >
> > There might be a misconception that a different node order should lead to a different generation sequence and thus the probability should be different. It is easy to have such a misconception from Figure 1 if you look at this Figure from left to right. However, in the generation procedure, we should consider the procedure from right to left. Let's consider two examples.
> >
> > 1) If the node order is (2, 3, 1, 4), which is shown in this figure, then the sampling procedure from right to left is just as the figure shows.
> > 2) However, if the node order is (3, 2, 1, 4), then the corresponding sampling procedure will first generate 1 and 4 in the same way as 1). In the third step, the generation of node 2 will have the same probability of generating node 3 in 1), that is, node 2 will play the role of node 3 in 1). Finally, the generation of node 3 will be the same as node 2 in 1). The generated graph will be the line graph (1, 3, 4, 2), which switches 2 and 3 in the graph shown in 1).
> >
> > In these two examples, we will have $p(G_{1:n}, \sigma_{1:n})$ exactly the same with either $\sigma_{1:n}=(2, 3, 1, 4)$ or $\sigma_{1:n}=(3, 2, 1, 4)$.
> >
> > Overall, I think the notation system of this paper is very confusing, and I spent many hours to figure out this issue. I would encourage the author to improve your notation system and consider these generation procedures as random events to study their relations (e.g. conditional probabilities).

---

> > > ### Author Response · Authors · 2022-12-06
> > > **The maginalization is incorrect**
> > >
> > > **Q1: According to your model decomposition, $p(G_{1:n},\sigma_{1:n})=p(G_{1:n-1}|\sigma_{1:n})p(\sigma_{1:n}|G_n)p(G_n)$. Since $p(G_{1:n-1}|\sigma_{1:n})$ is implemented by a GNN, which does not use node orders as the input, $p(G_{1:n-1}|\sigma_{1:n})$ is the same for any $\sigma_{1:n}$. Then by marginalizing out $\sigma_{1:n}$, we have $p(G_{1:n})=p(G_{1:n-1}|\sigma_{1:n})p(G_n)$. for an arbitrary $\sigma_{1:n}$. If you decide to use an uniform distribution $p(\sigma_{1:n}|G_n)=\frac{1}{n!}$ , then equation 8 should be $p(G_{1:n})=(n!)p(G_{1:n},\sigma_{1:n})$.**
> > >
> > > This is incorrect. $p(G_{1:n-1}|\sigma_{1:n})$ has the same nonzero value only for $\sigma_{1:n} \in \Pi(G_{1:n})$, where $\Pi(G_{1:n})$ is the set of all node orderings that give the same graph sequence $G_{1:n}$. However, the support of the generation ordering $p(\sigma_{1:n}|G_n)$ is over all $\sigma_{1:n}$. When $\sigma_{1:n} \notin \Pi(G_{1:n})$, the model cannot generate the graph sequence $G_{1:n}$ with the given node ordering and thus $p(G_{1:n-1}|\sigma_{1:n})=0$. The correct way to marginalize over $\sigma_{1:n}$ is thus:
> > > \begin{align}
> > > &\sum_{\sigma_{1:n}}p(G_{1:n-1}|\sigma_{1:n})p(\sigma_{1:n}|G_n)p(G_n)\\\\
> > > &=\sum_{\sigma_{1:n}\in \Pi(G_{1:n})} p(G_{1:n-1}|\sigma_{1:n})p(\sigma_{1:n}|G_n)p(G_n)+ \sum_{\sigma_{1:n}\notin \Pi(G_{1:n})} 0\cdot p(\sigma_{1:n}|G_n)p(G_n) \\\\
> > > &= \sum_{\sigma_{1:n}\in \Pi(G_{1:n})} p(G_{1:n-1}|\sigma_{1:n})p(\sigma_{1:n}|G_n)p(G_n) \\\\
> > > &=p(G_{1:n-1}|\sigma_{1:n})\sum_{\sigma_{1:n}\in \Pi(G_{1:n})} p(\sigma_{1:n}|G_n)p(G_n).
> > > \end{align}
> > >
> > > Since the support of $p(\sigma_{1:n}|G_n)$ is over all possible $\sigma_{1:n}$, $\sum_{\sigma_{1:n}\in \Pi(G_{1:n})} p(\sigma_{1:n}|G_n)\not=1$. Therefore, you cannot marginalize out $\sigma_{1:n}$ from $p(G_{1:n},\sigma_{1:n})=p(G_{1:n},\sigma_{1:n})p(\sigma_{1:n}|G_n)p(G_n)$ to obtain $p(G_{1:n})=p(G_{1:n-1}|\sigma_{1:n})p(G_n)$.
> > >
> > > How to compute $\sum_{\sigma_{1:n}\in \Pi(G_{1:n})} p(\sigma_{1:n}|G_n)$? This is a non-trivial problem since the generation ordering $p(\sigma_{1:n}|G_n)$ is not uniform and even has no closed-form expression. Instead, $p(\sigma_{1:n}|G_n)$ is separately modeled and set as the distribution of the diffusion ordering $q_{\phi}(\sigma_{1:n}|G_0)$ during training to achieve the optimal point of VLB (Full illustration can be found in our previous response https://openreview.net/forum?id=98J48HZXxd5&noteId=jdqpPlL6yG).
> > >
> > > **Q2: However, there is one question not answered: why Chen et al. also use the same equation as (8)? Is their equation wrong as well? After I studied their paper, I feel that your $p(G_{1:n},\sigma_{1:n})$ is different from theirs. If I assume your distribution $p(\sigma_{1:n}|G_n)$ is uniform, then $p(G_{1:n},\sigma_{1:n})$ is the same for all $\sigma_{1:n}$, but their $p(G_{1:n},\sigma_{1:n})$ does not have the behavior.**
> > >
> > > Our $p(G_{1:n},\sigma_{1:n})$ is exactly the same as Chen et al. As we illustrate in Q1, the marginalization is incorrect and you cannot assume $p(\sigma_{1:n}|G_n)$ is uniform. Therefore, our model does not have the behavior "$p(G_{1:n},\sigma_{1:n})$ is the same for all $\sigma_{1:n}$" either.
> > >
> > > **Q3: There might be a misconception that a different . ...In these two examples, we will have $p(G_{1:n},\sigma_{1:n})$ exactly the same with either  or $\sigma_{1:n}=(2,3,1,4)$ or $\sigma_{1:n}=(3,2,1,4)$.**
> > >
> > > We did consider that different node orderings can lead to the same graph sequence and the same probability $p(G_{1:n},\sigma_{1:n})$. That is exactly why our derivation involves the graph automorphism $\Pi(G_{1:n})$, i.e., the set of all node orderings that give the same graph sequence $G_{1:n}$.
> > >
> > > As we illustrate in our previous response (https://openreview.net/forum?id=98J48HZXxd5&noteId=-36LwSy98IA), the main difference of our method compared with OM is how to treat $p(G_{1:n},\sigma_{1:n})$. OM factorizes $p(G_{1:n},\sigma_{1:n})$ as
> > > $$p(G_{1:n},\sigma_{1:n})=p(G_{1:n})p(\sigma_{1:n}|G_{1:n}).$$
> > > As $p\left(\sigma_{1: n} \mid G_{1: n} \right)$ is a uniform distribution, OM inevitably needs to involve $|\mathit{\Pi}(G_{1:n})|$ in their VLB.
> > >
> > > In contrast, we factorize $p(G_{1: n}, \sigma_{1:n})$ as
> > > $$p\left(G_{1: n}, \sigma_{1: n}\right) = p\left(G_{1: n-1} \mid \sigma_{1: n}\right) p\left(\sigma_{1: n} \mid G_n\right) p\left(G_n\right),$$
> > > which has two benefits:
> > > (1) The challenge of modeling the node ordering variable is now pushed into the term $p\left(\sigma_{1: n} \mid G_n\right)$. This is a key reason that we do not have the $\left|\Pi\left(G_{1: n}\right)\right|$ term in our VLB, instead we have a KL term. Fortunately, we do not need to explicity model this distribution and simply set it as the diffusion ordering $q(\sigma_{1:n}|G_n)$ during training.
> > > (2) Our generator network $p_{\theta}(G_{0:n-1}|G_n,\sigma_{1:n})$ models the graph generation conditioned on a fixed node ordering.  In contrast, OM models $p(G_{1:n})$ with no exact node ordering information.

---

> > > > ### Comment · Reviewer_dB1j · 2022-12-06
> > > > **Can you redo the derivation use the adjacency matrix?**
> > > >
> > > > I have done a lot of work trying to convert your notation to the adjacency matrix. Your generation model in Section 3.2 essentially generates an adjacency matrix with a particular row order. I think a good practice is to follow notations established by previous work (GraphRNN, OM, and others). If you do so, then your deviation error becomes obvious.
> > > >
> > > > For example, you mention that $p(G_{1:n-1} | \sigma_{1:n}) = 0$ when $\sigma \notin \Pi(G_{1:n})$. However, with any node order $\sigma$, the generative model in section 3.2 can always generate a graph sequence $G_{1:n-1}$ but with nodes labeled according to $\sigma$.
> > > >
> > > > Furthermore, you may need to explain this observation: if we use $p(G_{1:n}) = (n!) p(G_{1:n}, \sigma_{1:n})$, then your variational lower bound match exactly to that of (Chen et al., 2021).

---

> > > > > ### Author Response · Authors · 2022-12-08
> > > > > **Adjacency matrix should lead to exactly the same deriving procedure.**
> > > > >
> > > > > **I have done a lot of work trying to convert your notation to the adjacency matrix. Your generation model in Section 3.2 essentially generates an adjacency matrix with a particular row order. I think a good practice is to follow notations established by previous work (GraphRNN, OM, and others). If you do so, then your derivation error becomes obvious.**
> > > > >
> > > > > We cannot use exactly the same notation of previous work (GraphRNN, OM). The previous works first use the node ordering $\sigma$ to permute the adjacency matrix and then autoregressively model the probability of the permuted adjacency matrix $p(A^{\sigma})$. In their notation, $A_{t}^{\sigma}$ represents the first $t$ rows of the permuted adjacency matrix $A^{\sigma}$. However, the permuted adjacency matrix $A^{\sigma}$ does not contain the exact node ordering information $\sigma$ since multiple node orders can lead to the same adjacency matrix. While in our model, we predict the edge connectivity of the exact node $v_{\sigma_t}$ at each step and thus generate/denoise the graph in the exact node order $\sigma$.
> > > > >
> > > > > An equivalent way to use adjacency matrix is to first fix the adjacency matrix $A$ with a random row order and assign each row with a node in the graph. Then we unmask the $\sigma_t$-th row of the fixed matrix $A$ at each step $t$, starting from the fully masked matrix $A_n$. $A_t$ represents rows $\sigma_i, i=t,\cdots,n$ are unmasked. **However, this is exactly the same as our graph notation** --- predicting the edge connectivity of node $v_{\sigma_t}$ in $G_n$ at each step. By replacing $G_t$ with $A_t$, we will obtain the same deriving procedure.
> > > > >
> > > > > Why can we use the exact node ordering information while previous work cannot? This is because, in our framework, the node ordering $\sigma_{1:n}$ is sampled in the forward diffusion process which essentially assigns a unique ID to each node of $G_0$. In the generation procedure, our starting point is the masked graph $G_n$ with $n$  nodes; our generator network sequentially denoises the node $v_{\sigma_t}$ in $G_n$ in the reverse order of the diffusion ordering during training. In contrast, the previous work can only sequentially grow the permuted adjacency matrix but such permuted matrix loses the exact node ordering information.
> > > > >
> > > > >
> > > > > **For example, you mention that $p(G_{1:n-1}|\sigma_{1:n})=0$ when $\sigma_{1:n} \notin \Pi(G_{1:n})$. However, with any node order $\sigma$, the generative model in section 3.2 can always generate a graph sequence $G_{1:n-1}$ but with nodes labeled according to $\sigma$.**
> > > > >
> > > > > Yes, every node ordering $\sigma_{1:n}$ can lead to a corresponding graph sequence $G_{1:n}$.
> > > > > However, during the marginalization, you need to integrate over all $\sigma_{1:n}$ for a particular graph sequence $G_{1:n}$. For $\sigma_{1:n}\notin \Pi(G_{1:n})$,
> > > > > $p(G_{1:n-1}|\sigma_{1:n})=0$ since this node ordering cannot lead to this particular graph sequence $G_{1:n}$.
> > > > >
> > > > >
> > > > > **Furthermore, you may need to explain this observation: if we use $p(G_{1:n})=(n!)p(G_{1:n},\sigma_{1:n})$, then your variational lower bound match exactly to that of (Chen et al., 2021).**
> > > > >
> > > > > First, as we illustrate before, your marginalization and your assumption about $p(\sigma_{1:n}|G_n)$ is  incorrect. Therefore, we cannot obtain this equation.
> > > > >
> > > > > Second, even this equation was correct, you still could not obtain the exact VLB in Chen et al. Your VLb does not need to compute $|\Pi(G_{1:n})|$ but $n!$.

---

> > > > > > ### Comment · Reviewer_dB1j · 2022-12-08
> > > > > > **The discussion is not productive if notations are not well-defined.**
> > > > > >
> > > > > > For example, what's the exact definition of $G_{1:n}$? Does $G_{1:n}$ has nodes labeled for each graph $G_i$? If so, how can you use the equation from (Chen et al., 2021); if not, then any $\sigma$ can get such a graph sequence.
> > > > > >
> > > > > > I think a more rigorous definition of notations is needed to show the correctness of the result. With the current notation system, it is hard to have a productive discussion. I will stop responding unless I see an exact derivation step I miss.

---

> > > > > > > ### Author Response · Authors · 2022-12-11
> > > > > > > **G_{1:n} is defined by definition 3 and 4**
> > > > > > >
> > > > > > > The node of $G_{1:n}$ is not labeled according to our definitions of absorbing node state (definition 3) and autoregressive graph diffusion process (definition 4).
> > > > > > >
> > > > > > >
> > > > > > > You cannot label $G_{1:n}$ in any order $\sigma_{1:n}$ for $p(G_{0:n-1}|\sigma_{1:n})$ because the node ID in $p(G_{0:n-1}|\sigma_{1:n})$ must be consistent with the node ID in the diffusion ordering $q(\sigma_{1:n}|G_0)$. Only when they are consistent, you can compute the expectation $p(G_{0:n-1}|\sigma_{1:n})$ under the diffusion ordering $q(\sigma_{1:n}|G_0)$.
> > > > > > >
> > > > > > >
> > > > > > > We do not need to label $G_{1:n}$ since we have the corresponding diffusion process for every generation process during training. The forward diffusion process assigns a unique ID to each node. Then, at each step of the reverse generation process, we first select which unique node to unmask and then add the **connectivity of the selected node** to the graph structure $G_t$.

---

### Official Review · Reviewer_uUZJ · 2022-10-25

**Confidence:** 4
**Clarity, Quality, Novelty And Reproducibility:** Clarity, Quality, Novelty And Reprodu…
**Correctness:** 4
**Technical Novelty And Significance:** 4
**Empirical Novelty And Significance:** 3
**Recommendation:** 6

**Strength And Weaknesses:**

### Strength
- This paper gives a new method for the graph generation.
- The well-designed model obtains better performance.

### Weaknesses
- It does not well explain what autoregressive diffusion is.
- Theoretical analysis is not sufficient

**Summary Of The Paper:**

- This paper proposed an autoregressive diffusion-based graph generation model.
- Its performance outperforms SOTAs.
- GRAPHARM also has low complexity.

**Summary Of The Review:**

- This paper gives a new method for the graph generation.
- The well-designed model obtains better performance.

---

> ### Author Response · Authors · 2022-11-19
> **Reply to Reviewer uUZJ's Comments**
>
> **Q1. It does not well explain what autoregressive diffusion is.**
>
> We have a detailed introduction of the autoregressive diffusion process in the Background section. We also introduce the connection between autoregressive diffusion and absorbing diffusion with continuous time limit in Appendix A.2.
>
> **Q2. Theoretical analysis is not sufficient**
>
> We have added a detailed derivation of our variational lower bound based on the theory of graph automorphism in Appendix A.3.

---

### Official Review · Reviewer_EBGH · 2022-10-25

**Confidence:** 3
**Correctness:** 3
**Technical Novelty And Significance:** 2
**Empirical Novelty And Significance:** 2
**Recommendation:** 6

**Clarity, Quality, Novelty And Reproducibility:**

This paper is clearly written. The proposed autoregressive diffusion model is new, but the inference algorithm, especially the node ordering inference part, is similar to Chen et al. 2021.

**Strength And Weaknesses:**

1. This paper introduces a discrete autoregressive diffusion process on graphs that is exchangeable in node ordering. The number of steps required is proportional to the number of nodes, thus improving sampling efficiency over previous graph sampling methods.

2. The paper resolves the node ordering inference problem involved in the inference process by exploiting the variational lower bound property. The method is similar to "Order matters: probabilistic modeling of node sequence for graph generation." by (Chen et al. 2021).

3. Empirical results demonstrate superior performance in generation quality and time complexity.

Weaknesses:
1. The paper does not clearly explain the "optimal node generation ordering" learned from the posterior. An example demonstrated in Figure 3 in the appendix shows that GraphRAM tends to generate one community and then generate the other community. But more insight is required to check why that ordering is preferred in GraphRAM.

**Summary Of The Paper:**

This paper proposes an autoregressive diffusion model (ARM) that leverages the absorbing discrete diffusion as the diffusion process on graphs. Since node ordering is involved in the diffusion process, the generative process needs to learn the reversed node ordering for the graph generation. The paper exploits the objective function in the KL divergence and uses a reinforcement learning procedure to bypass the inference complexity. Empirical results demonstrate superior performance compared with baseline methods.

**Summary Of The Review:**

This paper proposes a diffusion process on discrete graphs that derives an autoregressive generative process with inferred node ordering. Empirical results are good, but my concern is its similarity with previous node ordering inference algorithms.

---

> ### Author Response · Authors · 2022-11-19
> **Reply to Reviewer EBGH's Comments**
>
> **Q1. The paper does not clearly explain the "optimal node generation ordering" learned from the posterior. An example demonstrated in Figure 3 in the appendix shows that GraphARM tends to generate one community and then generate the other community. But more insight is required to check why that ordering is preferred in GraphARM.**
>
> The optimal node generation ordering should reflect the graph topology/regularity.
> We quantitatively evaluate the generation ordering of GRAPHARM on the community-small dataset and compare it with OA-ARM which uses a random generation ordering. We first generate 100 graphs using both methods and record the node generation orderings. Then, we use the spectral graph clustering method to partition the nodes into two clusters for each generated graph. Finally, we count the nodes that cross different clusters during the generation procedure. As we can see from Table 1, the average number of nodes that cross different clusters of our method is much smaller than OA-ARM. This demonstrates that our method tends first to generate one cluster and then add another cluster, while OA-ARM just randomly generates the graph. Therefore, the generation ordering of GRAPHARM can reflect the graph topology/regularity better than OA-ARM.
>
> |Method | Average Counts |
> |----|---|
> | GraphARM| 2.6|
> |OA-ARM|6.9|
>
> Table 1. Average number of nodes that cross different clusters on the community-small dataset.
>
> **Q2. My concern is its similarity with previous node ordering inference algorithms**
>
> Our method is different from OM [1] in two aspects:
>
> (1) The motivations are different. Our method is motivated by autoregressive diffusion  and treats the graph sequences in the diffusion process as the latent variable; OM treats the node ordering as the latent variable and infers its posterior distribution in a way similar to Variational autoencoder (VAE). (2) Our training objective is much simpler than OM. First, we do not need to compute the complicated graph automorphism, which requires some approximation algorithms to compute. Second, our training objective does not involve the entropy of the node ordering distribution. Existing works [2, 3] have shown that the entropy term in the VAE objective can easily cause posterior collapse. That is a key reason why our method can consistently outperform OM, as shown in our experiments. We have added this experiment in our updated manuscript.
>
> We have also added a discussion about the differences between our method and OM in Section 3.5 and a detailed derivation of our variational lower bound in Appendix A.3.
>
> ### References
> [1] Chen, Xiaohui, et al, Order Matters: Probabilistic Modeling of Node Sequence for Graph Generation, *ICML 2021*.
>
> [2] Lucas, James, et al, Understanding Posterior Collapse in Generative Latent Variable Models, *ICLR 2019 Workshop*.
>
> [3] Lucas, James, et al, Don't Blame the ELBO! A Linear VAE Perspective on Posterior Collapse, *NeurIPS 2019*.

---

### Author Response · Authors · 2022-11-19
**Summary of the Changes to the Revised Manuscript**

Dear Reviewers,

Thank you all for your comments! We are grateful for the feedback and have improved the manuscript. We have posted a revision of the paper with changes highlighted in red. Our main changes are as follows:
1. We modify the description of the diffusion process as suggested by review dB1j.
2. Add a subsection to discuss the difference between our method and OM [1] in Section 3.5.
3. We highlight that our method can consistently outperform OM because we do not suffer from posterior collapse in the experimental results section.
4. We quantitatively evaluate the node generation ordering in Table 4.
5. We add a detailed derivation of our variational lower bound (Eq.3) in Appendix A.3.
6. We empirically evaluate the KL-divergence between the diffusion ordering and the generation ordering in Eq.3 indeed approaches to 0 in Appendix A.9.


[1] Chen, Xiaohui, et al, Order Matters: Probabilistic Modeling of Node Sequence for Graph Generation, *ICML 2021*.

---

### Decision · Program_Chairs · 2023-01-20

**Decision:**

Reject

**Justification For Why Not Higher Score:**

See the summary of the Summary Of AC-reviewer Meeting.

**Justification For Why Not Lower Score:**

See the summary of the Summary Of AC-reviewer Meeting.

**Metareview: Summary, Strengths And Weaknesses:**

This paper proposes an autoregressive diffusion model (GraphArm)  for graph generation, based on the standard diffusion model paradigm with two separate forward and reverse diffusion processes. During training, the forward process incrementally masks nodes of the graph, connecting each masked node with all the other nodes. The forward process thus ends with a full-connected graph of masked nodes. A “denoising network” is then trained to recover the original graph, starting from the node masked last.

A key design choice in the forward process is how to set the order of nodes being masked. Instead of uniform sampling of possible orders, the authors propose a “diffusing ordering network” to learn the optimal order. This diffusion ordering network is jointly trained with the denoising network using a log-likelihood objective. Experiments on six different graphs suggest that GraphArm outperforms or matches strong baselines.

The reviewers agreed that the paper is well-written (reviewers EBHG and KoQP), the results are promising (reviewers EBHG, dB1j, and KoQP), and the proposed method has certain novelty (reviewers dB1j and KoQP). On the other hand, some reviewers (EBHG and KoQP) pointed out the similarities of the proposed method to previous work (Chen et al., 2021).  Also, serious concerns were raised and debated intensively by a reviewer (dB1j) on the correctness of some proofs/formula in this paper.

During the rebuttal, the authors addressed these concerns with extensive discussions, and submitted a revised version with additional theoretical analysis. The AC carefully checked those discussions, and recommended a (weak) rejection, or a conditional acceptance if no other borderline papers looked stronger.



**Summary Of Ac-Reviewer Meeting:**

The AC called for zoom meetings (twice) and had email conversation's with the reviewers who could not attend the meetings.  The main (or the only) focus in those meetings/conversations is regarding the correctness of the proofs/formula in this paper, as raised by reviewer dB1.  All the other reviewers did not express any clear options in one way or another on the debated proofs; some of them expressed an OK with either acceptance or rejection.

The AC carefully read the debate between a reviewer (dB1j) and the authors on the correctness of some derivations/formulas in this paper, and have the following observations:

1) The reviewer thinks that the VLB in Eq.3 of this paper is the same as that in OM (Chen et al, 2021). But, the authors point out the difference as "we treat the graph sequences in the diffusion process as the latent variables while OM treats the node ordering as the latent variable" and "We believe this is a key reason why our method consistently outperforms OM empirically."

2) The reviewer thinks that Eq.8 in this paper$$p_\theta(G_{1:n}, \sigma_{1:n}) = \frac{1}{|\Pi(G_{1:n})|} p_\theta (G_{1:n})$$ is incorrect, and that the correct formula should be (assuming a uniform distribution of node orderings) $$p_\theta(G_{1:n}, \sigma_{1:n}) = \frac{1}{n!} p_\theta (G_{1:n})$$  However, the authors point out that the uniform assumption is incorrect because only a subset of the node orderings (denoted as $\Pi(G_{1:n})$) gives the same graph sequence $G_{1:n}$ In other words, when $\sigma_{1:n}\not \in \Pi(G_{1:n})$, the model cannot generate the graph sequence $G_{1:n}$ with the given node ordering and hence we have $$p_\theta(G_{1:n}| \sigma_{1:n}) =0$$

The AC found the authors' statements to be more on the correct side. Nonetheless, given the mediocre scores of 6 or less and the lack of excitement among the reviewers, the AC feels that the paper is incremental by nature, and hence suggest a rejection but would not fight against an acceptance if the program chairs were decided to do so.